# STRIVING FOR SIMPLICITY IN OFF-POLICY DEEP REINFORCEMENT LEARNING

## ABSTRACT

This paper advocates the use of offline (batch) reinforcement learning (RL) to help (1) isolate the contributions of exploitation vs. exploration in off-policy deep RL, (2) improve reproducibility of deep RL research, and (3) facilitate the design of simpler deep RL algorithms. We demonstrate that recent off-policy deep RL algorithms, even when trained solely on all of the replay data of an online DQN agent, can outperform online DQN on Atari 2600 games. We also present Random Ensemble Mixture (REM), a simple $Q$-learning algorithm that enforces optimal Bellman consistency on random convex combinations of multiple $Q$-value estimates. Using the DQN replay dataset, offline REM surpasses the gains from online C51 and outperforms offline QR-DQN. Furthermore, REM performs comparably to QR-DQN in the online RL setting on Atari 2600 games. The DQN replay dataset can serve as an offline RL benchmark and will be released.

## 1 INTRODUCTION

Deep neural networks have become a critical component of modern reinforcement learning (RL) (Sutton and Barto, 2018). The seminal work of Mnih et al. (2013; 2015) on deep $Q$-networks (DQN) has demonstrated that it is possible to train neural networks using $Q$-learning (Watkins and Dayan, 1992) to achieve human-level performance in playing Atari 2600 games (Bellemare et al., 2013) directly from raw pixels. Recent progress in mastering Go (Silver et al., 2016) and advances in robotic control (Levine et al., 2016; OpenAI et al., 2018; Kalashnikov et al., 2018) present additional supporting evidence for the enormous potential of deep RL.

Off-policy RL algorithms such as $Q$-learning are attractive because they disentangle data collection and policy optimization and offer more sample efficient solutions than on-policy algorithms (Sutton et al., 2000; Schulman et al., 2015; Mnih et al., 2016). Importantly, off-policy techniques can leverage the vast amount of existing offline logged data for real-world applications such as digital advertising (Strehl et al., 2010; Bottou et al., 2013), education (Mandel et al., 2014), and healthcare (Shortreed et al., 2011). Since online RL is often unsafe to deploy in the real world, offline RL algorithms are the only feasible solution for many practical decision making problems (Dulac-Arnold et al., 2019). Nevertheless, off-policy RL algorithms when combined with neural networks can be unstable or even divergent (Baird, 1995; Boyan and Moore, 1995; Tsitsiklis and Van Roy, 1997).

In the absence of theoretical guarantees for off-policy deep RL, recent advances (see Hessel et al. (2018) for an overview) are largely governed by empirical results on a popular benchmark suite of Atari 2600 games (Bellemare et al., 2013). Unfortunately, these empirical results are difficult to reproduce (Henderson et al., 2018) and the contribution of novel RL algorithms is often conflated with many other design choices (Clary et al., 2019; Khetarpal et al., 2018). Given the empirical nature of deep off-policy RL, it is crucial to come up with simpler and reproducible experimental settings and study the relative importance of different components of deep RL algorithms. It is also important to strive for finding successful RL algorithms that are as simple as possible.

This paper advocates the use of offline (batch) RL to help (1) isolate the contributions of *exploitation vs. exploration* in off-policy deep RL, (2) improve reproducibility of deep RL research, and (3) facilitate the design of simpler off-policy deep RL algorithms. To this end, we propose a new benchmark for offline optimization of RL agents on Atari 2600 games using all of the replay data of a DQN agent (Mnih et al., 2015). Using this benchmark, we investigate the following questions:

1. Is it possible to train successful Atari agents based solely on offline data?
2. Can one design simple and effective alternatives to intricate RL algorithms in the offline setting?
3. Are the insights gained from the offline setting useful for developing effective online algorithms?

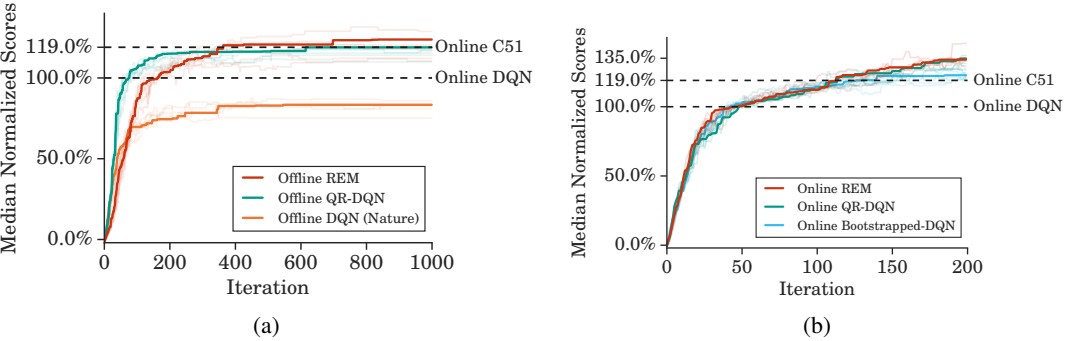

Figure 1: Median normalized online evaluation scores averaged over 5 runs (shown as traces) across stochastic version of 60 Atari 2600 games of (a) **offline** RL agents trained using the DQN replay dataset containing 200 million frames. Offline REM outperforms offline QR-DQN and surpasses gains from online C51. (b) **online** RL agents trained for 200 million frames (standard protocol). Online REM with 4 $Q$-networks performs comparably to online QR-DQN. Each iteration corresponds to 1 million frames.

The contributions of this paper can be summarized as:

- An offline RL benchmark is proposed for evaluating and designing RL algorithms on Atari 2600 games without exploration, based on the logged replay data of a DQN agent comprising 50 million (observation, action, reward, next observation) tuples per game. This reduces the computation cost of the experiments considerably and helps improve reproducibility of deep RL research by standardizing training using a fixed offline dataset. The replay dataset used in our experiments will be released to enable offline optimization of RL algorithms on a common ground.

- Contrary to recent work (Zhang and Sutton, 2017; Fujimoto et al., 2019), we find that the logged DQN data is sufficient for optimizing strong Atari agents offline without any environment interaction. For instance, QR-DQN (Dabney et al., 2018b) trained offline on the DQN replay dataset significantly outperforms online DQN (Mnih et al., 2015).

- A simple and novel $Q$-learning algorithm called *Random Ensemble Mixture (REM)* is presented, which enforces optimal Bellman consistency on random convex combinations of multiple $Q$-value estimates. Offline REM, trained using the DQN replay dataset, outperforms more complex RL algorithms such as offline QR-DQN and surpasses the gains from online C51 (Bellemare et al., 2017) over online DQN (Figure 1a).

- We use the insights gained from the offline experiments to develop an online variant of REM, which performs comparably with online QR-DQN (Figure 1b).

## 2 OFF-POLICY REINFORCEMENT LEARNING

An interactive environment in reinforcement learning (RL) is typically described as a Markov decision process (MDP) $(\mathcal{S}, \mathcal{A}, R, P, \gamma)$ (Puterman, 1994), with a state space $\mathcal{S}$, an action space $\mathcal{A}$, a stochastic reward function $R(s, a)$, transition dynamics $P(s'|s, a)$ and a discount factor $\gamma \in [0, 1)$. A stochastic policy $\pi(\cdot \mid s)$ maps each state $s \in \mathcal{S}$ to a distribution (density) over actions.

For an agent following the policy $\pi$, the action-value function, denoted $Q^\pi(s, a)$, is defined as the expectation of cumulative discounted future rewards, *i.e.,*

$$Q^\pi(s, a) = \mathbb{E}\left[\sum\nolimits_{t=0}^{\infty} \gamma^t r_t \,\Big|\, s_0 = s, a_0 = a, s_t \sim P(\cdot \mid s_{t-1}, a_{t-1}), a_t \sim \pi(\cdot \mid s_t), r_t \sim R(s_t, a_t)\right]. \tag{1}$$

The goal of RL is to find an optimal policy $\pi^*$ that attains maximum expected return, for which $Q^{\pi^*}(s, a) \geq Q^\pi(s, a)$ for all $\pi, s, a$. The Bellman optimality equations (Bellman, 1957) characterize the optimal policy in terms of the optimal $Q$-values, denoted $Q^* = Q^{\pi^*}$, via:

$$Q^*(s, a) = \mathbb{E}\left[r + \gamma \max_{a'} Q^*(s', a') \,\Big|\, r \sim R(s, a),\ s' \sim P(\cdot \mid s, a)\right]. \tag{2}$$

To learn a policy from interaction with the environment, $Q$-learning (Watkins and Dayan, 1992) iteratively improves an approximate estimate of $Q^*$, denoted $Q_\theta$, by repeatedly regressing the LHS of (2) to target values defined by samples from the RHS of (2). For large and complex state

spaces, approximate $Q$-values are obtained using a neural network as the function approximator. To further stabilize optimization, a target network $Q_{\theta'}$ with frozen parameters may be used for computing the learning target (Mnih et al., 2013). The target network parameters $\theta'$ are updated to the current $Q$-network parameters $\theta$ after a fixed number of time steps. DQN (Mnih et al., 2013; 2015) parameterizes $Q_\theta$ with a convolutional neural network (LeCun et al., 1998) and uses $Q$-learning with a target network while following an $\epsilon$-greedy policy with respect to $Q_\theta$ for data collection. DQN minimizes the TD error using the loss $\mathcal{L}(\theta)$ on mini-batches of agent's past experience tuples, $(s, a, r, s')$, sampled from an experience replay buffer $\mathcal{D}$ (Lin, 1992) collected during training:

$$\mathcal{L}(\theta) \; = \; \mathbb{E}_{s,a,r,s' \sim \mathcal{D}} \left[ \ell_\delta \left( Q_\theta(s,a) - r - \gamma \max_{a'} Q_{\theta'}(s',a') \right) \right], \tag{3}$$

where $l_\delta$ is the Huber loss (Huber, 1964) given by $\ell_\delta(u) = \begin{cases} \frac{1}{2}u^2 & \text{for } |u| \le \delta, \\ \delta(|u| - \frac{1}{2}\delta), & \text{otherwise.} \end{cases}$.

$Q$-learning is an *off-policy* algorithm (Sutton and Barto, 2018) since the learning target can be computed without any consideration of how the experience was generated. In offline (batch) RL (Ernst et al., 2005; Riedmiller, 2005; Lange et al., 2012), we assume access to a fixed offline dataset of experiences $\mathcal{D}$, without any further interaction with the environment.

A family of recent off-policy deep RL algorithms, which serve as a strong baseline in this paper, include Distributional RL (Bellemare et al., 2017; Jaquette, 1973) methods. Such algorithms estimate a density over returns for each state-action pair, denoted $Z^\pi(s, a)$, instead of directly estimating the mean $Q^\pi(s, a)$. Accordingly, one can express a form of distributional Bellman optimality as

$$Z^*(s,a) \overset{D}{=} r + \gamma Z^*(s', \mathrm{argmax}_{a' \in \mathcal{A}} Q^*(s',a')), \qquad \text{where } r \sim R(s,a), \; s' \sim P(\cdot \mid s,a), \tag{4}$$

and $\overset{D}{=}$ denotes distributional equivalence and $Q^*(s', a')$ is estimated by taking an expectation with respect to $Z^*(s', a')$. C51 (Bellemare et al., 2017) approximates $Z^*(s, a)$ by using a categorical distribution over a set of pre-specified anchor points, and distributional QR-DQN (Dabney et al., 2018b) approximates the return density by using a uniform mixture of $K$ Dirac delta functions, *i.e.,*

$$Z_\theta(s,a) := \frac{1}{K} \sum_{i=1}^{K} \delta_{\theta_i(s,a)}, \quad \text{and} \quad Q_\theta(s,a) = \frac{1}{K} \sum_{i=1}^{K} \theta_i(s,a) \,. \tag{5}$$

QR-DQN trains each $\theta_i$ for $1 \le i \le K$ to match the $\frac{2i-1}{2K}$-quantile of the target return density (RHS of (4)) using the Huber quantile regression loss (Koenker, 2005; Dabney et al., 2018b). QR-DQN, albeit complex, outperforms C51 and DQN and obtains state-of-the-art results on Atari 2600 games, among agents that do not exploit $n$-step updates (Sutton, 1988) and prioritized replay (Schaul et al., 2016). This paper avoids using $n$-step updates and prioritized replay to keep the empirical study simple and focused on deep $Q$-learning algorithms.

## 3 OFFLINE OPTIMIZATION OF REINFORCEMENT LEARNING AGENTS

Modern off-policy deep RL algorithms perform remarkably well on common benchmarks such as the Atari 2600 games (Bellemare et al., 2013) and continuous control MuJoCo tasks (Todorov et al., 2012). Such off-policy algorithms (Mnih et al., 2015; Lillicrap et al., 2015) are considered "online" because they alternate between optimizing a policy and using that policy to collect more data. Typically, these algorithms keep a sliding window of most recent experiences in a finite replay buffer (Lin, 1992), throwing away stale data to incorporate most fresh (on-policy) experiences. In principle, off-policy algorithms can learn from data collected by any policy, however, Zhang and Sutton (2017) assert that using a large replay buffer can significantly hurt the performance of $Q$-learning algorithms, since it can delay rare on-policy experiences that are important for policy learning.

This paper revisits offline off-policy RL and investigates whether off-policy deep RL agents trained solely on offline data can be successful. We advocate the use of offline RL to help isolate an RL algorithm's ability to *exploit* experience and generalize *vs.* its ability to *explore* effectively by collecting interesting new data. The offline RL setting removes design choices related to the replay buffer and exploration; therefore, it is much simpler to experiment with and reproduce than the typical online off-policy learning. Also, in the offline setting, we optimize an objective (*e.g.,* (3)) over a fixed dataset as opposed to a changing replay buffer of an online agent. In this work, we mainly experiment

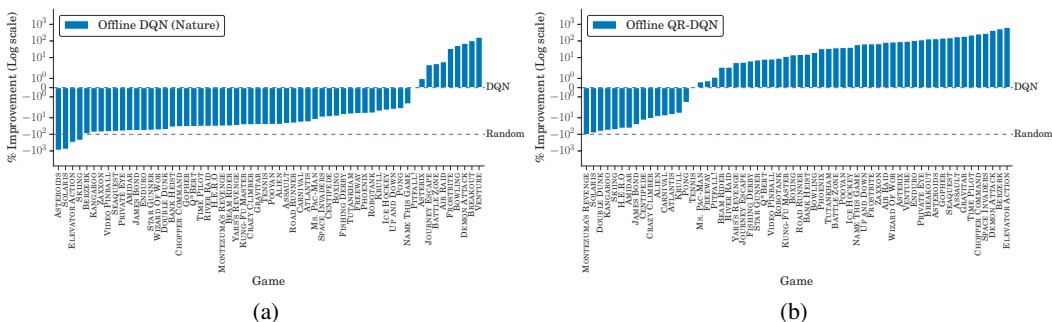

(a)                                                        (b)

Figure 2: Normalized performance improvement (in %) over online DQN (Nature), per game, of (a) offline DQN (Nature) and (b) offline QR-DQN trained using the DQN replay dataset for same number of gradient updates as online DQN. The normalized online score for each game is 0.0 and 1.0 for the worse and better performing agent among fully trained online DQN and random agents respectively.

with large and diverse datasets motivated by real-world RL problems where we already have access to (usually continually growing) diverse datasets of logged experiences, *e.g.,* historical data from recommender systems (Bottou et al., 2013), database of robotic experiences (Dasari et al., 2019) *etc*.

To facilitate a study of offline RL with large and diverse datasets, we train several instances of DQN (Nature) agents (Mnih et al., 2015) on stochastic version of 60 Atari 2600 games for 200 million frames each, with a frame skip of 4 (standard protocol) and sticky actions enabled (Machado et al., 2018). On each game, we train 5 different agents with random initialization, and store all of the tuples of (observation, action, reward, next observation) encountered during training into 5 replay datasets of 50 million tuples each. These replay datasets are used for training standard off-policy RL agents, offline, without any interaction with the environment during training. We perform an online evaluation of the agents in the intervals of 1 million training frames, and report the best evaluation score for each agent, averaged over 5 runs.

## 3.1 EXPERIMENTS AND RESULTS

Given the logged replay data of the DQN agent, it is natural to ask whether training an offline variant of DQN on this data will recover the performance of online DQN. Furthermore, whether more recent off-policy algorithms are able to exploit the DQN replay dataset more effectively than DQN. To investigate these questions, we train DQN (Nature) and distributional QR-DQN (Dabney et al., 2018b) agents, offline, on the DQN replay dataset for the same number of gradient updates as online DQN. We use the hyperparameters provided in the Dopamine baselines (Castro et al., 2018) for a standardized comparison (Section A.4) and report scores using a normalized scale (Section A.3). We note that the replay datasets include samples from all of the intermediate policies seen during the optimization of the DQN (Nature) agent. Accordingly, we compare the performance of offline agents against the best performing policy among the mixture of data collecting policies.

**Results.** We find that offline DQN underperforms online DQN on all except a few games while offline QR-DQN outperforms offline DQN and online DQN on most of the games (Figure 2). C51 (Bellemare et al., 2017) trained offline using the DQN replay dataset also considerably improves upon offline DQN (Figure A.4). These results suggest that DQN (Nature) is somewhat ineffective at exploiting off-policy data. However, offline QR-DQN's impressive performance demonstrates that it is possible to optimize strong Atari agents completely offline (see Figure A.6 for learning curves).

## 3.2 OFFLINE CONTINUOUS CONTROL EXPERIMENTS

In contrast to our offline results on Atari 2600 games with discrete action spaces, discussed above, Fujimoto et al. (2019) find that standard off-policy deep RL agents are not effective on continuous control problems when trained offline, even when large and diverse replay datasets are used. The results of Fujimoto et al. (2019) are based on the evaluation of a standard continuous control agent, called DDPG (Lillicrap et al., 2015), and other more recent continuous control algorithms such as TD3 (Fujimoto et al., 2018) and SAC (Haarnoja et al., 2018) are not considered in their study.

Motivated by the so-called *final buffer* setting in Fujimoto et al. (2019) (Section A.2), we train a DDPG agent on continuous control MuJoCo tasks (Todorov et al., 2012) for 1 million time steps and store all of the experienced transitions. Using this dataset, we train standard off-policy agents including

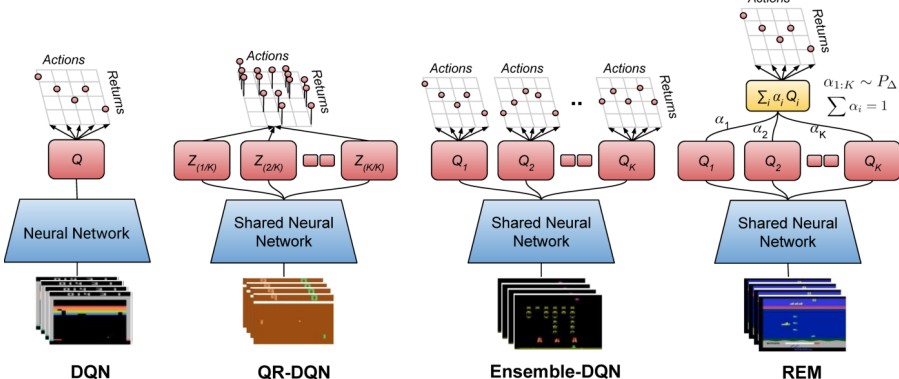

Figure 3: Neural network architectures for DQN, QR-DQN and the proposed expected RL variants, *i.e.,* Ensemble-DQN and REM, with the same multi-head architecture as QR-DQN. The individual $Q$-heads share all of the neural network layers except the final fully connected layer.

TD3 and DDPG completely offline. Consistent with our offline results on Atari games, offline TD3 significantly outperforms the data collecting DDPG agent and offline DDPG (Figure A.1).

## 4 SEEKING SIMPLER OFF-POLICY RL ALGORITHMS

Section 3.1 finds that recent distributional RL algorithms such as QR-DQN are effective at exploiting offline data. However, these algorithms are complicated as they require a generalization of Bellman equations to distributions (4) and typically minimize a probability divergence measure from a distributional target (Rowland et al., 2018; Dabney et al., 2018b) instead of scalar TD errors. Moreover, despite recent efforts (Lyle et al., 2019; Bellemare et al., 2019), the source of the gains when using distributional RL algorithms remains unclear. We investigate whether one can develop simple and effective alternatives to more intricate RL algorithms in the offline setting. To this end, we study two deep $Q$-learning algorithms, Ensemble DQN and REM, which adopt ensembling to improve stability.

### 4.1 ENSEMBLE-DQN

Ensemble-DQN is a simple extension of DQN that approximates the $Q$-values via an ensemble of parameterized $Q$-functions (Faußer and Schwenker, 2015; Osband et al., 2016; Anschel et al., 2017). Each $Q$-value estimate, denoted $Q^k_\theta(s, a)$, is trained against its own target $Q^k_{\theta'}(s, a)$, similar to Bootstrapped-DQN (Osband et al., 2016). The $Q$-functions are optimized using identical mini-batches in the same order, starting from different parameter initializations. The loss $\mathcal{L}(\theta)$ takes the form,

$$\mathcal{L}(\theta) \ = \ \frac{1}{K} \sum_{k=1}^{K} \mathbb{E}_{s,a,r,s' \sim \mathcal{D}} \left[ \ell_\delta \left( Q^k_\theta(s, a) - r - \gamma \max_{a'} Q^k_{\theta'}(s', a') \right) \right] , \qquad (6)$$

where $l_\delta$ is the Huber loss. While Bootstrapped-DQN uses one of the $Q$-value estimates in each episode to improve exploration, in the offline setting, we are only concerned with the ability of Ensemble-DQN to exploit better and use the mean of the $Q$-value estimates for evaluation.

### 4.2 RANDOM ENSEMBLE MIXTURE (REM)

Increasing the number of models used for ensembling typically improves the performance of supervised learning models (Shazeer et al., 2017). This raises the question whether one can use an ensemble over an exponential number of $Q$-estimates in a computationally efficient manner. Inspired by dropout (Srivastava et al., 2014), we propose Random Ensemble Mixture for RL.

Random Ensemble Mixture (REM) uses multiple parameterized $Q$-functions to estimate the $Q$-values, similar to Ensemble-DQN. The key insight behind REM is that one can think of a convex combination of multiple $Q$-value estimates as a valid $Q$-value estimate itself. This is especially true at the fixed point, where all of the $Q$-value estimates have converged to an identical $Q$-function. Using this insight, we train a family of $Q$-function approximators defined by mixing probabilities on a $(K - 1)$-simplex.

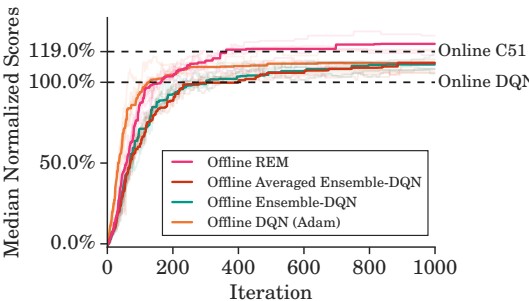

Figure 4: **Offline REM *vs*. Baselines**. Normalized scores averaged over 5 runs (shown as traces) across 60 stochastic Atari 2600 games of offline agents trained using the DQN replay dataset trained for 5X gradient steps.

Table 1: Median normalized best average evaluation scores (averaged over 5 runs) across 60 stochastic Atari 2600 games, measured as percentages and number of games where an offline agent achieves better scores than a fully trained online DQN (Nature) agent.

| Offline agent | Median | $>$ DQN |
|---|---|---|
| DQN (Adam) | 111.9% | 41 |
| Ensemble-DQN | 111.0% | 39 |
| Averaged Ensemble-DQN | 112.1% | 43 |
| QR-DQN | 118.9% | 45 |
| REM | **123.8%** | **49** |

Specifically, for each mini-batch, we randomly draw a categorical distribution $\alpha$, which defines a convex combination of the $K$ estimates to approximate the optimal $Q$-function. This approximator is trained against its corresponding target to minimize the TD error. The loss $\mathcal{L}(\theta)$ takes the form,

$$\mathcal{L}(\theta) = \mathbb{E}_{s,a,r,s'\sim\mathcal{D}}\left[\mathbb{E}_{\alpha_1,\ldots,\alpha_K\sim P_\Delta}\left[\ell_\delta\left(\sum_k \alpha_k Q_\theta^k(s,a) - r - \gamma\max_{a'}\sum_k \alpha_k Q_{\theta'}^k(s',a')\right)\right]\right] \tag{7}$$

where $P_\Delta$ represents a probability distribution over the standard $(K-1)$-simplex $\Delta^{K-1} = \{\alpha \in \mathbb{R}^K : \alpha_1 + \alpha_2 + \cdots + \alpha_K = 1, \alpha_k \geq 0, k = 1, \ldots, K\}$.

REM considers $Q$-learning as a constraint satisfaction problem based on Bellman optimality constraints (2) and $\mathcal{L}(\theta)$ can be viewed as an infinite set of constraints corresponding to different mixture probability distributions. For action selection, we use the average of the $K$ value estimates as the $Q$-function, *i.e.*, $Q(s,a) = \sum_k Q_\theta^k(s,a)/K$. REM is easy to implement and analyze (see Proposition 1), and can be viewed as a simple regularization technique for value-based RL. In our experiments, we use a very simple distribution $P_\Delta$: we first draw a set of $K$ values *i. i. d.* from Uniform (0, 1) and normalize them to get a valid categorical distribution, *i.e.*, $\alpha_k' \sim U(0,1)$ followed by $\alpha_k = \alpha_k'/\sum \alpha_i'$.

**Proposition 1**. *Consider the assumptions: (a) The distribution $P_\Delta$ has full support over the entire $(K-1)$-simplex. (b) Only a finite number of distinct $Q$-functions globally minimize the loss in (3). (c) $Q^*$ is defined in terms of the MDP induced by the data distribution $\mathcal{D}$. (d) $Q^*$ lies in the family of our function approximation. Then, at the global minimum of $\mathcal{L}(\theta)$ (7) for a multi-head $Q$-network:*

   *(i) Under assumptions (a) and (b), all the $Q$-heads represent identical $Q$-functions.*
   *(ii) Under assumptions (a)–(d), the common global solution is $Q^*$.*

The proof of *(ii)* follows from *(i)* and the fact that (7) is lower bounded by the TD error attained by $Q^*$. The proof of part *(i)* can be found in the supplementary material.

### 4.3 EXPERIMENTS AND RESULTS

**Asymptotic performance of offline agents.** In supervised learning, asymptotic performance matters much more than performance within a fixed budget of gradient updates. Similarly, for a given sample complexity, we prefer RL algorithms that perform the best as long as the number of gradient updates is feasible. Since the sample efficiency of the offline agents for a given dataset is fixed, we train them on the DQN replay dataset for five times as many gradient updates as online DQN.

**Comparison with QR-DQN.** QR-DQN modifies the DQN (Nature) architecture to output $K$ values for each action using a multi-head $Q$-network and replaces RMSProp (Tieleman and Hinton, 2012) with Adam (Kingma and Ba, 2015) for optimization. To ensure a fair comparison with QR-DQN, we use the same multi-head $Q$-network as QR-DQN with $K = 200$ heads (Figure 3), where each head represents a $Q$-value estimate for REM and Ensemble-DQN. We also use Adam for optimization.

**Additional Baselines.** To isolate the gains due to Adam in QR-DQN and our proposed variants, we compare against a DQN baseline which also uses Adam. We also evaluate Averaged Ensemble-DQN, a variant of Ensemble-DQN proposed by Anschel et al. (2017), which uses the average of the predicted target $Q$-values as the Bellman target for training each parameterized $Q$-function. This

Figure 5: **Effect of Dataset Size.** Normalized scores (averaged over 5 runs) of QR-DQN and multi-head REM trained offline on stochastic version of 5 Atari 2600 games for 5X gradient steps using only a fraction of the entire DQN replay dataset (200M frames) obtained via randomly subsampling trajectories.

baseline determines whether the random combinations of REM provide any significant benefit over simply using an ensemble of predictors to stabilize the Bellman target.

**REM with separate $Q$-networks.** To investigate whether the effectiveness of REM is linked to ensembling techniques and dropout, we compare offline REM with $K$ $Q$-value estimates computed using $K$ separate $Q$-networks *vs.* a multi-head $Q$-network for $K \in \{4, 16\}$.

**Results.** Figure 4a shows the comparison of the additional baselines with REM and Ensemble-DQN and Table 1 summarizes the asymptotic performance results. DQN with Adam noticeably bridges the gap in asymptotic performance between QR-DQN and DQN (Nature) in the offline setting. Offline Ensemble-DQN does not improve upon this strong DQN baseline showing that its naive ensembling approach is inadequate. Furthermore, Averaged Ensemble-DQN performs only slightly better than Ensemble-DQN. In contrast, REM exploits offline data more effectively than other agents, including QR-DQN, when trained for more gradient updates. Surprisingly, using the DQN replay dataset, offline REM surpasses the gains from online C51, a huge improvement over online DQN (Figure 1a).

Based on above empirical results, we hypothesize that gains from REM are due to the noise from randomly ensembling $Q$-value estimates leading to more robust training analogous to dropout. Furthermore, REM with separate $Q$-networks performs better asymptotically and learns faster than REM with a multi-head $Q$-network in the offline setting (Figure A.5). This consolidates our belief about effectiveness of REM being linked to dropout as separate $Q$-networks result in more variation in individual $Q$-estimates leading to more robust training.

### 4.4 FROM OFFLINE TO ONLINE REINFORCEMENT LEARNING

Can we combine the insights gained from the offline setting with appropriate design choices (*e.g.,* exploration, replay buffer) to create effective online methods? In online RL, learning and data generation are tightly coupled, *i.e.,* an agent that learns faster also collects more relevant training data.

We ran online REM with $K$ separate $Q$-networks (with $K = 4$ for computational efficiency) because of the improved convergence speed over multi-head REM in the offline setting. For data collection, we use $\epsilon$-greedy with a randomly sampled $Q$-estimate from the simplex for each episode, similar to Bootstrapped DQN. To estimate the gains from the REM objective (7) in the online setting, we also evaluate Bootstrapped-DQN with identical modifications (*e.g.,* separate $Q$-networks) as online REM. Figure 1b show that REM performs on par with QR-DQN and considerably outperforms Bootstrapped-DQN (see Figure A.8 for learning curves). Based on these results, we conclude that the offline setting can help facilitate the design of effective online off-policy RL algorithms.

### 5 OFFLINE RL: DATASET SIZE AND COMPOSITION

Our offline learning results indicate that 50 million logged experience tuples per game from DQN (Nature) are sufficient to obtain good online performance on most of the Atari 2600 games. One may argue that the size of the fixed replay and its composition play a key role in our empirical results (De Bruin et al., 2015). To study the role of the replay buffer size in the success of the offline agents, we perform an ablation experiment with variable replay buffer size.

We conducted offline experiments with reduced data obtained via randomly subsampling entire trajectories from the logged DQN experiences, thereby maintaining the same data distribution. Figure 5 presents the performance of the offline REM and QR-DQN agents with $N\%$ of the tuples in the DQN replay dataset where $N \in \{1, 10, 20, 50, 100\}$. As expected, performance tends to increase

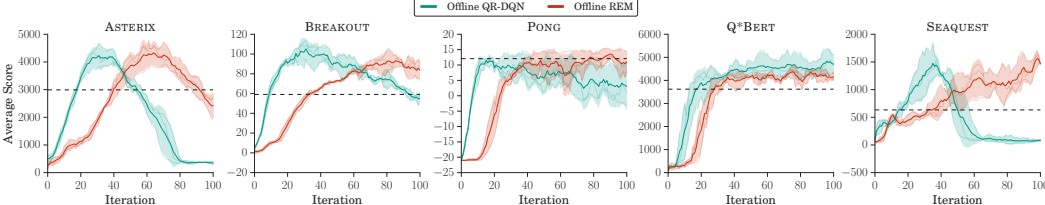

Figure 6: **Effect of Dataset Quality.** Normalized scores (averaged over 3 runs) of QR-DQN and multi-head REM trained offline on stochastic version of 5 Atari 2600 games for 5X gradient steps using logged data from online DQN trained only for 20M frames (20 iterations). The horizontal line shows the performance of best policy found during DQN training for 20M frames which is significantly worse than fully-trained DQN.

as the fraction of data increases. With $N \geq 10\%$, REM and QR-DQN still perform comparably to online DQN on most of these games. However, the performance deteriorates drastically for $N = 1\%$.

To see the effect of quality of offline dataset, we perform another ablation where we train offline REM and QR-DQN on the first 20 million frames in the DQN replay dataset which approximate exploration data with suboptimal returns. Similar to our offline results with the entire replay dataset, on most Atari games, offline REM and QR-DQN perform comparably and outperform the best policy amongst the mixture of policies which collected the first 20 million frames (Figure 6 and Figure A.9).

## 6  DISCUSSION AND FUTURE WORK

Since the *(observation, action, next observation, reward)* tuples in DQN replay dataset are stored in the order they were experienced by online DQN during training, various data collection strategies for *benchmarking* offline RL can be induced by subsampling the replay dataset containing 200 million frames. For example, the first $k$ million frames from the DQN replay dataset emulate exploration data with suboptimal returns (*e.g.,* Figure 6) while the last $k$ million frames are analogous to near-expert data with stochasticity. Another option is to randomly subsample the entire dataset to create smaller offline datasets with sufficient diversity (*e.g.,* Figure 5). Based on the popularity and ease of experimentation on Atari games, we hope that the DQN replay dataset can be used for benchmarking offline RL in addition to continuous control setups Fujimoto et al. (2019); Kumar et al. (2019).

Offline RL tackles the challenge of squeezing the most performance gains out of a fixed batch of data. This indicates the utility of offline RL for creating more *sample-efficient* RL agents as an online agent should exploit the data collected prior to any given time during online training as much as possible before further collecting new data via exploration. The proposed offline benchmark can be utilized for developing better RL algorithms for various data regimes shown in Figure 5.

In our offline RL setup, the agents exhibit overfitting on some games (Figure 6), even with 50 million tuples (Figure A.7), *i.e.,* after a sufficiently large number of gradient updates, their performance degrades. We also observe divergence w.r.t. Huber loss in our reduced data experiments with $N = 1\%$. Based on these findings, we posit that our offline benchmark can be used for designing techniques akin to regularization for *stable* off-policy deep RL algorithms with reasonable performance throughout learning (García and Fernández, 2015), a requirement for many practical RL applications.

Finally, investigating Rainbow components, *e.g.,* prioritized replay (Schaul et al., 2016), $n$-step updates (Sutton, 1988), double $Q$-learning (Van Hasselt et al., 2016), *etc.,* in the offline RL setting and incorporating them with REM are interesting directions. Experience replay-based algorithms can be more sample efficient than model-based approaches (Van Hasselt et al., 2019), and using the DQN replay dataset on Atari 2600 games for designing non-parametric replay models (Pan et al., 2018) and parametric world models (Kaiser et al., 2019) is quite promising for improving sample-efficiency. We also leave further investigation of the exploitation ability of distributional RL to future work.

## 7  RELATED WORK

Our work is motivated by whether the complexity of recent off-policy approaches (*e.g.,* Rainbow (Hessel et al., 2018)) is necessary and is similar in spirit to work by Rajeswaran et al. (2017), which attempts to demystify the complexity of on-policy deep RL for continuous control. Our work is mainly related to two subareas of RL known as batch RL and distributional RL. For the sake of clarity, we refer to batch RL as offline RL in other sections of the paper.

**Batch Reinforcement Learning**. While there has been increasing interest in batch RL (Lange et al., 2012) over the last few years (Jiang and Li, 2016; Farajtabar et al., 2018; Irpan et al., 2019), much of this focussed on off-policy policy evaluation, where the goal is to estimate the performance of a given fixed policy. Similar to (Ernst et al., 2005; Kalyanakrishnan and Stone, 2007; Jaques et al., 2019), we investigate batch off-policy learning, which requires learning a good policy given a fixed dataset. In our offline setup, we only assume access to samples from the behavior policy and focus on $Q$-learning methods without any importance sampling correction as opposed to (Swaminathan and Joachims, 2015; Liu et al., 2019). Recent work (Fujimoto et al., 2019; Kumar et al., 2019) documents that standard off-policy methods with fixed datasets fail on continuous control MuJoCo tasks. Opposed to Kumar et al. (2019), we focus on the offline setting with data collected from a diverse mixture of policies rather than a single Markovian behavior policy. Furthermore, in contrast to Fujimoto et al. (2019), our batch learning results on MuJoCo tasks and Atari 2600 games (Section 3) demonstrate that standard off-policy deep RL algorithms (*e.g.,* TD3 (Fujimoto et al., 2018), QR-DQN (Dabney et al., 2018b)) are quite effective when learning truly off-policy from large and diverse datasets. Zhang and Sutton (2017) show that large replay buffers can hurt the performance of simple $Q$-learning methods with weak function approximators; they attribute this to the "off-policyness" of a large buffer which might delay important transitions for learning. However, our results reveal that even with uniform sampling, effective deep $Q$-learning algorithms (*e.g.,* REM) can exploit completely off-policy logged DQN data on Atari 2600 games, given sufficient number of gradient updates.

**Distributional Reinforcement Learning**. Recently, distributional RL algorithms (*e.g.,* C51 (Bellemare et al., 2017), QR-DQN (Dabney et al., 2018b), IQN (Dabney et al., 2018a)) gained popularity due to their strong performance on Atari 2600 games. We observe that distributional algorithms (*e.g.,* QR-DQN, C51) are quite effective at exploitation and develop much simpler REM (Section 4.2), which is as successful as QR-DQN. In contrast to distributional RL, much of the theoretical results of $Q$-learning naturally extends to REM due to its similarity to $Q$-learning, *e.g.,* in the tabular setting, REM converges to $Q^*$, while popular distributional methods (*e.g.,* QR-DQN, C51) need not learn true return statistics corresponding to the underlying MDP (Rowland et al., 2019).

## 8 CONCLUSIONS

This work investigates off-policy deep RL using an offline RL benchmark on Atari 2600 games based on logged experiences of a DQN agent. We demonstrate that it is possible to learn policies with high returns that significantly outperform the policies used to collect replay data. We develop REM, a simple variant of ensemble $Q$-learning, that can effectively exploit large-scale off-policy data and shares much of the success of the more complicated distributional RL algorithms on Atari 2600 games. The proposed benchmark can serve as a testbed for designing simple, stable and sample-efficient RL algorithms and enable the research community to evaluate off-policy methods on a common ground. As mentioned by Sutton and Barto (2018, Chapter 11.10): "*The potential for off-policy learning remains tantalizing, the best way to achieve it still a mystery.*"

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

# A APPENDIX

## A.1 PROOFS

**Proposition 1**. *Consider the assumptions: (a) The distribution* $P_\Delta$ *has full support over the entire* $(K - 1)$*-simplex. (b) Only a finite number of distinct Q-functions globally minimize the loss in* (3). *(c)* $Q^*$ *is defined in terms of the MDP induced by the data distribution* $\mathcal{D}$. *(d)* $Q^*$ *lies in the family of our function approximation. Then at the global minimum of* $\mathcal{L}(\theta)$ (7) *for multi-head Q-network :*

*(i) Under assumptions (a) and (b), all the Q-heads represent identical Q-functions.*
*(ii) Under assumptions (a)–(d), the common convergence point is* $Q^*$.

**Proof**. Part *(i)*: Under assumptions (a) and (b), we would prove by contradiction that each $Q$-head should be identical to minimize the REM loss $\mathcal{L}(\theta)$ (7). Note that we consider two $Q$-functions to be distinct only if they differ on any state $s$ in $\mathcal{D}$.

The REM loss $\mathcal{L}(\theta) = \mathbb{E}_{\alpha \sim P\Delta} [\mathcal{L}(\alpha, \theta)]$ where $\mathcal{L}(\alpha, \theta)$ is given by

$$\mathcal{L}(\alpha, \theta) = \mathbb{E}_{s,a,r,s' \sim \mathcal{D}} \left[ \ell_\delta \left( \sum_k \alpha_k Q_\theta^k(s, a) - r - \gamma \max_{a'} \sum_k \alpha_k Q_{\theta'}^k(s', a') \right) \right]. \quad (8)$$

If the heads $Q_\theta^i$ and $Q_\theta^j$ don't converge to identical $Q$-values at the global minimum of $\mathcal{L}(\theta)$, it can be deduced using Lemma 1 that all the $Q$-functions given by the convex combination $\alpha_i Q_\theta^i + \alpha_j Q_\theta^j$ such that $\alpha_i + \alpha_j = 1$ minimizes the loss in (3). This contradicts the assumption that only a finite number of distinct $Q$-functions globally minimize the loss in (3). Hence, all $Q$-heads represent an identical $Q$-function at the global minimum of $\mathcal{L}(\theta)$.

**Lemma 1**. *Assuming that the distribution* $P_\Delta$ *has full support over the entire* $(K - 1)$*-simplex* $\Delta^{K-1}$, *then at any global minimum of* $\mathcal{L}(\theta)$, *the Q-function heads* $Q_\theta^k$ *for* $k = 1, \ldots, K$ *minimize* $\mathcal{L}(\alpha, \theta)$ *for any* $\alpha \in \Delta^{K-1}$.

**Proof**. Let $Q_{\alpha^*, \theta^*} = \sum_{k=1}^K \alpha_k^* Q_{\theta^*}^k(s, a)$ corresponding to the convex combination $\alpha^* = (\alpha_1^*, \cdots, \alpha_K^*)$ represents one of the global minima of $\mathcal{L}(\alpha, \theta)$ (8) *i.e.,* $\mathcal{L}(\alpha^*, \theta^*) = \min_{\alpha, \theta} \mathcal{L}(\alpha, \theta)$ where $\alpha \in \Delta^{K-1}$. Any global minima of $\mathcal{L}(\theta)$ attains a value of $\mathcal{L}(\alpha^*, \theta^*)$ or higher since,

$$\mathcal{L}(\theta) = \mathbb{E}_{\alpha \sim P\Delta} [\mathcal{L}(\alpha, \theta)] \geq \mathbb{E}_{\alpha \sim P\Delta} [\mathcal{L}(\alpha^*, \theta^*)] \geq \mathcal{L}(\alpha^*, \theta^*) \quad (9)$$

Let $Q_{\theta^*}^k(s, a) = w_{\theta^*}^k \cdot f_{\theta^*}(s, a)$ where $f_{\theta^*}(s, a) \in \mathbb{R}^D$ represent the shared features among the $Q$-heads and $w_{\theta^*}^k \in \mathbb{R}^D$ represent the weight vector in the final layer corresponding to the $k$-th head. Note that $Q_{\alpha^*, \theta^*}$ can also be represented by each of the individual $Q$-heads using a weight vector given by convex combination $\alpha^*$ of weight vectors $(w_{\theta^*}^1, \cdots, w_{\theta^*}^K)$, *i.e.,* $Q(s, a) = \left( \sum_{k=1}^K \alpha_k^* w_{\theta^*}^k \right) \cdot f_{\theta^*}(s, a)$.

Let $\theta^I$ be such that $Q_{\theta^I}^k = Q_{\alpha^*, \theta^*}$ for all $Q$-heads. By definition of $Q_{\alpha^*, \theta^*}$, for all $\alpha \sim P_\Delta$, $\mathcal{L}(\alpha, \theta^I) = \mathcal{L}(\alpha^*, \theta^*)$ which implies that $\mathcal{L}(\theta^I) = \mathcal{L}(\alpha^*, \theta^*)$. Hence, $\theta^I$ corresponds to one of the global minima of $\mathcal{L}(\theta)$ and any global minima of $\mathcal{L}(\theta)$ attains a value of $\mathcal{L}(\alpha^*, \theta^*)$.

Since $\mathcal{L}(\alpha, \theta) \geq \mathcal{L}(\alpha^*, \theta^*)$ for any $\alpha \in \Delta^{K-1}$, for any $\theta^M$ such that $\mathcal{L}(\theta^M) = \mathcal{L}(\alpha^*, \theta^*)$ implies that $\mathcal{L}(\alpha, \theta^M) = \mathcal{L}(\alpha^*, \theta^*)$ for any $\alpha \sim P_\Delta$. Therefore, at any global minimum of $\mathcal{L}(\theta)$, the $Q$-function heads $Q_\theta^k$ for $k = 1, \ldots, K$ minimize $\mathcal{L}(\alpha, \theta)$ for any $\alpha \in \Delta^{K-1}$.

## A.2 OFFLINE CONTINUOUS CONTROL EXPERIMENTS

We replicated the *final buffer* setup as described by Fujimoto et al. (2019): We train a DDPG (Lillicrap et al., 2015) agent for 1 million time steps three standard MuJoCo continuous control environments in OpenAI gym (Todorov et al., 2012; Brockman et al., 2016), adding $\mathcal{N}(0, 0.5)$ Gaussian noise to actions for high exploration, and store all experienced transitions. This collection procedure creates a dataset with a diverse set of states and actions, with the aim of sufficient coverage. Similar to Fujimoto et al. (2019), we train DDPG across 15 seeds, and select the 5 top performing seeds for dataset collection.

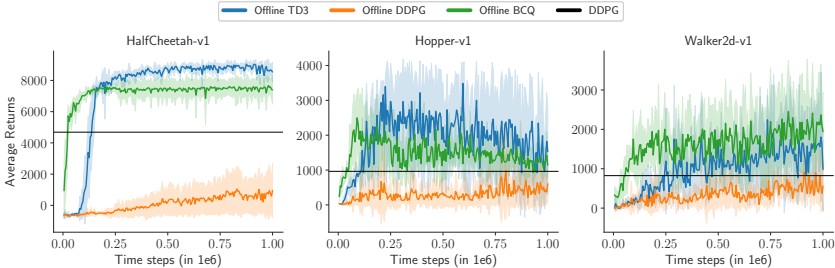

Figure A.1: We examine the performance of DDPG, TD3 and BCQ trained using identical offline data on three standard MuJoCo environments. We plot the mean performance (evaluated without exploration noise) across 5 runs. The shaded area represents a standard deviation. The bold black line measures the average return of episodes contained in the offline data collected using the DDPG agent (with exploration noise).

Using this logged dataset, we train standard continuous control off-policy actor-critic methods namely DDPG and TD3 (Fujimoto et al., 2018) completely offline without any exploration. We also train a Batch-Constrained deep $Q$-learning (BCQ) agent, proposed by Fujimoto et al. (2019), which restricts the action space to force the offline agent towards behaving close to on-policy w.r.t. a subset of the given data. We use the open source code generously provided by the authors at `https://github.com/sfujim/BCQ` and `https://github.com/sfujim/TD3`. We use the hyperparameters mentioned in (Fujimoto et al., 2018; 2019) except offline TD3 which uses a learning rate of 0.0005 for both the actor and critic.

Figure A.1 show that offline TD3 significantly outperforms the behavior policy which collected the offline data as well as the offline DDPG agent. Noticeably, offline TD3 also performs comparably to BCQ, an algorithm designed specifically to learn from arbitrary, fixed offline data. While Fujimoto et al. (2019) attribute the failure to learn in the offline setting to extrapolation error (*i.e.,* the mismatch between the offline dataset and true state-action visitation of the current policy), our results suggest that failure to learn from diverse offline data may be linked to extrapolation error for only weak exploitation agents such as DDPG.

### A.3 SCORE NORMALIZATION

The improvement in normalized performance of an offline agent, expressed as a percentage, over an online DQN (Nature) (Mnih et al., 2015) agent is calculated as: $100 \times (\text{Score}_{\text{Normalized}} - 1)$ where:

$$\text{Score}_{\text{Normalized}} = \frac{\text{Score}_{\text{Agent}} - \min(\text{Score}_{\text{DQN}}, \text{Score}_{\text{Random}})}{\max(\text{Score}_{\text{DQN}}, \text{Score}_{\text{Random}}) - \min(\text{Score}_{\text{DQN}}, \text{Score}_{\text{Random}})}. \quad (10)$$

Here, the scores are the mean evaluation scores averaged over 5 runs. We chose not to measure performance in terms of percentage of online DQN scores alone because a tiny difference relative to the random agent on some games can translate into hundreds of percent in DQN score difference. Additionally, the max is needed since DQN performs worse than a random agent on the games Solaris and Skiing.

### A.4 HYPERPARAMETERS & EXPERIMENT DETAILS

In our experiments, we used the hyperparameters provided in Dopamine baselines (Castro et al., 2018) and report them for completeness and ease of reproducibility in Table 2. As mentioned by Dopamine's GitHub repository, changing these parameters can significantly affect performance, without necessarily being indicative of an algorithmic difference. We will also open source our code to further aid in reproducing our results.

The Atari environments (Bellemare et al., 2013) used in our experiments are stochastic due to sticky actions (Machado et al., 2018), *i.e.,* there is 25% chance at every time step that the environment will execute the agent's previous action again, instead of the agent's new action. All agents (online or offline) are compared using the best evaluation score (averaged over 5 runs) achieved during training where the evaluation is done online every training iteration using a $\epsilon$-greedy policy with $\epsilon = 0.001$. We report offline training results with same hyperparameters over 5 random seeds of the DQN replay data collection, game simulator and network initialization.

Table 2: The hyperparameters used by the offline and online RL agents in our experiments.

| Hyperparameter | setting (for both variations) |
|---|---|
| Sticky actions | Yes |
| Sticky action probability | 0.25 |
| Grey-scaling | True |
| Observation down-sampling | (84, 84) |
| Frames stacked | 4 |
| Frame skip (Action repetitions) | 4 |
| Reward clipping | [-1, 1] |
| Terminal condition | Game Over |
| Max frames per episode | 108K |
| Discount factor | 0.99 |
| Mini-batch size | 32 |
| Target network update period | every 2000 updates |
| Training steps per iteration | 250K |
| Update period every | 4 steps |
| Evaluation $\epsilon$ | 0.001 |
| Evaluation steps per iteration | 125K |
| $Q$-network: channels | 32, 64, 64 |
| $Q$-network: filter size | $8 \times 8, 4 \times 4, 3 \times 3$ |
| $Q$-network: stride | 4, 2, 1 |
| $Q$-network: hidden units | 512 |
| Multi-head $Q$-network: number of $Q$-heads | 200 |
| Hardware | Tesla P100 GPU |

| Hyperparameter | Online | Offline |
|---|---|---|
| Min replay size for sampling | 20,000 | - |
| Training $\epsilon$ (for $\epsilon$-greedy exploration) | 0.01 | - |
| $\epsilon$-decay schedule | 250K steps | - |
| Fixed Replay Memory | No | Yes |
| Replay Memory size | 1,000,000 steps | 50,000,000 steps |
| Replay Scheme | Uniform | Uniform |
| Training Iterations | 200 | 200 or 1000 |

**DQN replay dataset collection**. For collecting the offline data used in our experiments, we use online DQN (Nature) (Mnih et al., 2015) with the RMSprop (Tieleman and Hinton, 2012) optimizer. The DQN replay dataset, $\mathcal{B}_{\mathrm{DQN}}$, consists of approximately 50 million experience tuples for each run per game corresponds to 200 million frames due to frame skipping of four, *i.e.,* repeating a selected action for four consecutive frames. Note that the total dataset size is approximately 15 billion tuples ( $50 \frac{\text{million tuples}}{\text{agent}} * 5 \frac{\text{agents}}{\text{game}} * 60$ games).

**Optimizer related hyperparameters**. For existing off-policy agents, step size and optimizer were taken as published. Offline DQN (Adam) and all the offline agents with multi-head $Q$-network (Figure 3) use the Adam optimizer (Kingma and Ba, 2015) with same hyperparameters as online QR-DQN (Dabney et al., 2018b) (lr = 0.00005, $\epsilon_{\mathrm{Adam}} = 0.01/32$). Note that scaling the loss has the same effect as inversely scaling $\epsilon_{\mathrm{Adam}}$ when using Adam.

**Online Agents**. For online REM shown in Figure 1b, we performed hyper-parameter tuning over $\epsilon_{\mathrm{Adam}}$ in (0.01/32, 0.005/32, 0.001/32) over 5 training games (Asterix, Breakout, Pong, Q*Bert, Seaquest) and evaluated on the full set of 60 Atari 2600 games using the best setting (lr = 0.00005, $\epsilon_{\mathrm{Adam}} = 0.001/32$). Online REM uses 4 $Q$-value estimates calculated using separate $Q$-networks where each network has the same architecture as originally used by online DQN (Nature). Similar to REM, our version of Bootstrapped-DQN also uses 4 separate $Q$-networks and Adam optimizer with identical hyperaparmeters (lr = 0.00005, $\epsilon_{\mathrm{Adam}} = 0.001/32$).

**Wall-clock time for offline experiments.** The offline experiments are approximately 3X faster than the online experiments for the same number of gradient steps on a P100 GPU. In Figure 2 and 4(a), the offline agents are trained for 5X gradient steps, thus, the experiments are 1.67X slower than running online DQN for 200 million frames (standard protocol). Furthermore, since the offline experiments do not require any data generation, using tricks from supervised learning such as using much larger batch sizes than 32 with TPUs / multiple GPUS would lead to a significant speed up.

## A.5 ADDITIONAL PLOTS & TABLES

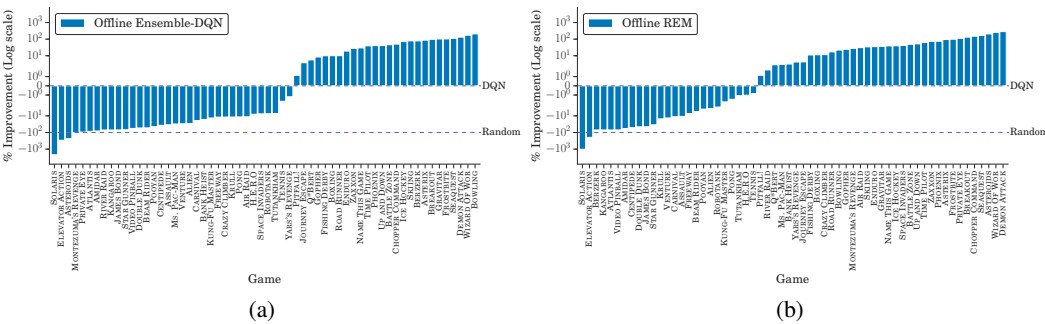

Figure A.2: Normalized Performance improvement (in %) over online DQN (Nature), per game, of (a) offline Ensemble-DQN and (b) offline REM trained using the DQN replay dataset for same number of gradient steps as online DQN. The normalized online score for each game is 0.0 and 1.0 for the worse and better performing agent among fully trained online DQN and random agents respectively.

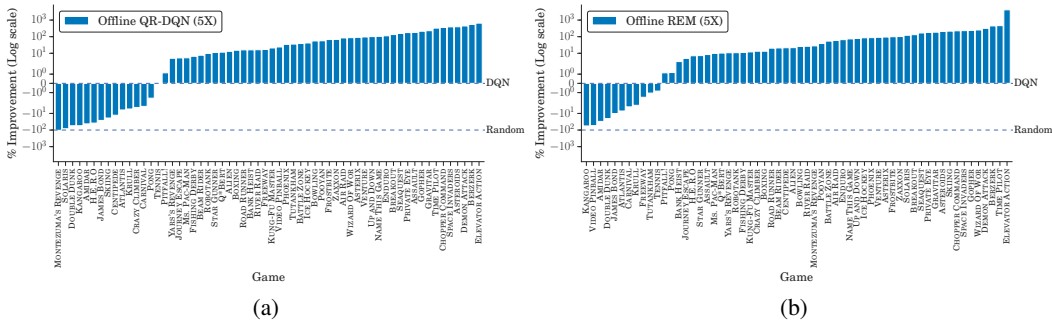

Figure A.3: Normalized Performance improvement (in %) over online DQN (Nature), per game, of (a) offline QR-DQN (5X) (b) offline REM (5X) trained using the DQN replay dataset for five times as many gradient steps as online DQN. The normalized online score for each game is 0.0 and 1.0 for the worse and better performing agent among fully trained online DQN and random agents respectively.

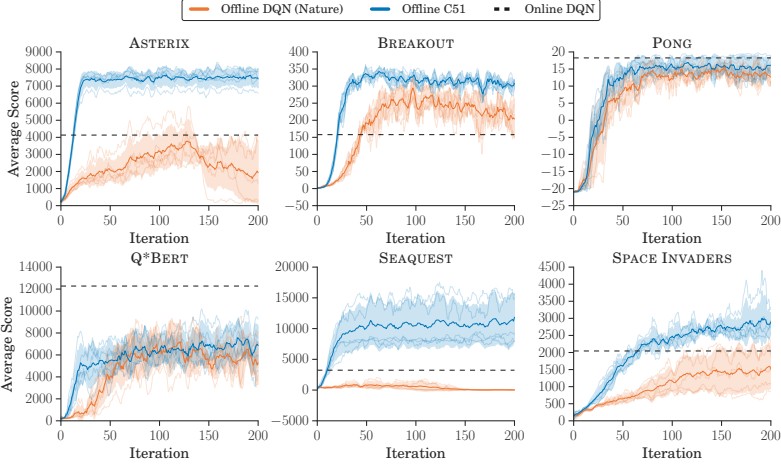

Figure A.4: **Offline C51 vs Offline DQN (Nature).** Average online scores of C51 and DQN (Nature) agents trained offline on stochastic version of 6 Atari 2600 games using the DQN replay dataset for the same number of gradient steps as the online DQN agent. The scores are averaged over 5 runs (shown as traces) and smoothed over a sliding window of 5 iterations and error bands show standard deviation.

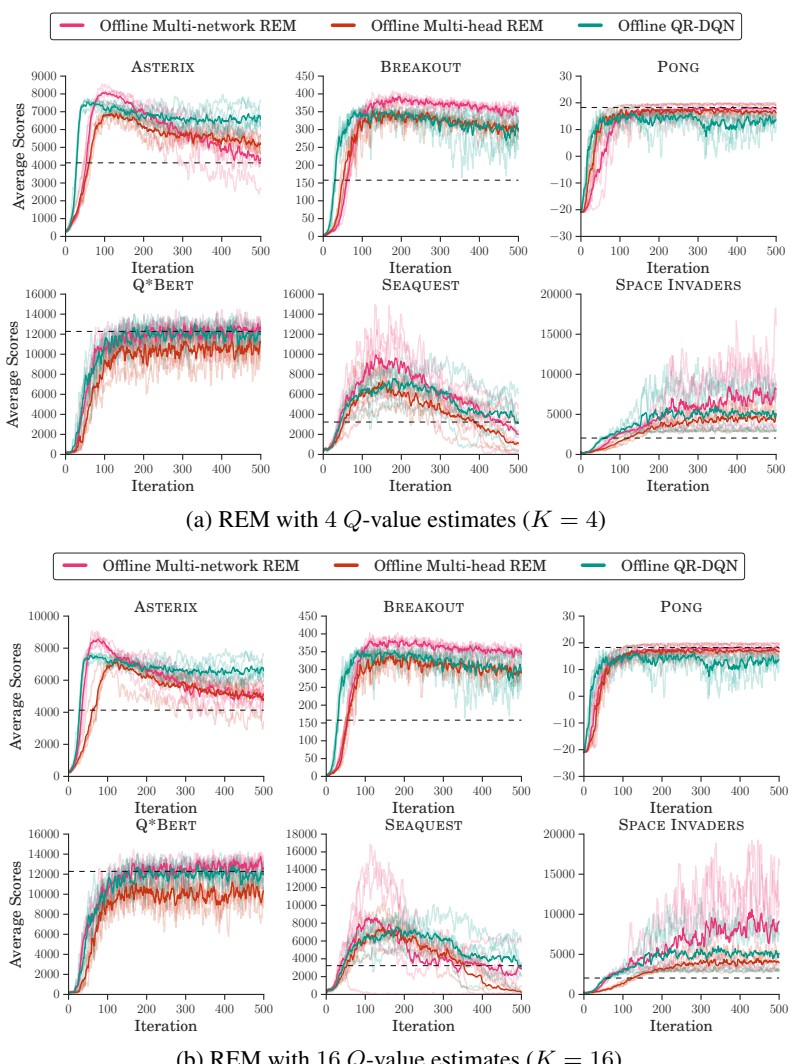

(a) REM with 4 $Q$-value estimates ($K = 4$)

(b) REM with 16 $Q$-value estimates ($K = 16$)

Figure A.5: **REM with Separate $Q$-networks.** Average online scores of offline REM variants with different architectures and QR-DQN trained on stochastic version of 6 Atari 2600 games for 500 iterations using the DQN replay dataset. The scores are averaged over 5 runs (shown as traces) and smoothed over a sliding window of 5 iterations and error bands show standard deviation. The multi-network REM and the multi-head REM employ $K$ $Q$-value estimates computed using separate $Q$-networks and $Q$-heads of a multi-head $Q$-network respectively and are optimized with identical hyperparameters. Multi-network REM improves upon the multi-head REM indicating that the more diverse $Q$-estimates provided by the separate $Q$-networks improve performance of REM over $Q$-estimates provided by the multi-head $Q$-network with shared features.

Table 3: Median normalized scores (Section A.3) across stochastic version of 60 Atari 2600 games, measured as percentages and number of games where an agent achieves better scores than a fully trained online DQN (Nature) agent. All the offline agents below are trained using the DQN replay dataset. The entries of the table without any suffix report training results with the five times as many gradient steps as online DQN while the entires with suffix ($1x$) indicates the same number of gradient steps as the online DQN agent. All the offline agents except DQN use the same multi-head architecture as QR-DQN.

| Offline agent | Median | > DQN | Offline agent | Median | > DQN |
|---|---|---|---|---|---|
| DQN (Nature) ($1x$) | 74.4% | 10 | DQN (Nature) | 83.4% | 17 |
| DQN (Adam) ($1x$) | 104.6% | 39 | DQN (Adam) | 111.9% | 41 |
| Ensemble-DQN ($1x$) | 92.5% | 26 | Ensemble-DQN | 111.0% | 39 |
| Averaged Ensemble-DQN ($1x$) | 88.6% | 24 | Averaged Ensemble-DQN | 112.1% | 43 |
| QR-DQN ($1x$) | **115.0**% | **44** | QR-DQN | 118.9% | 45 |
| REM ($1x$) | 103.7% | 35 | REM | **123.8**% | **49** |

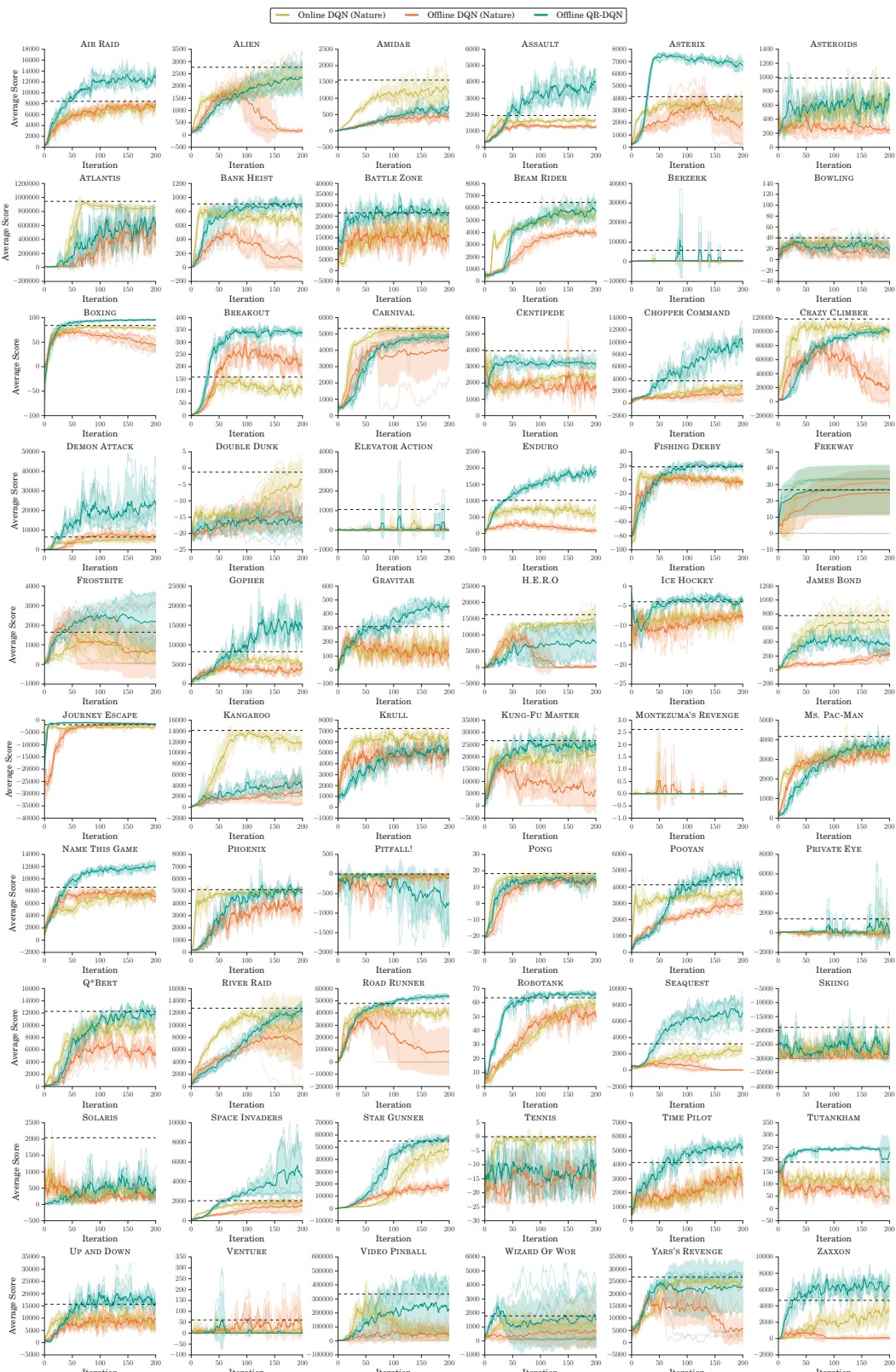

Figure A.6: Average evaluation scores across stochastic version of 60 Atari 2600 games for online DQN, offline DQN and offline QR-DQN trained for 200 iterations. The offline agents are trained using the DQN replay dataset. The scores are averaged over 5 runs (shown as traces) and smoothed over a sliding window of 5 iterations and error bands show standard deviation. The horizontal line shows the performance of the best policy (averaged over 5 runs) found during training of online DQN.

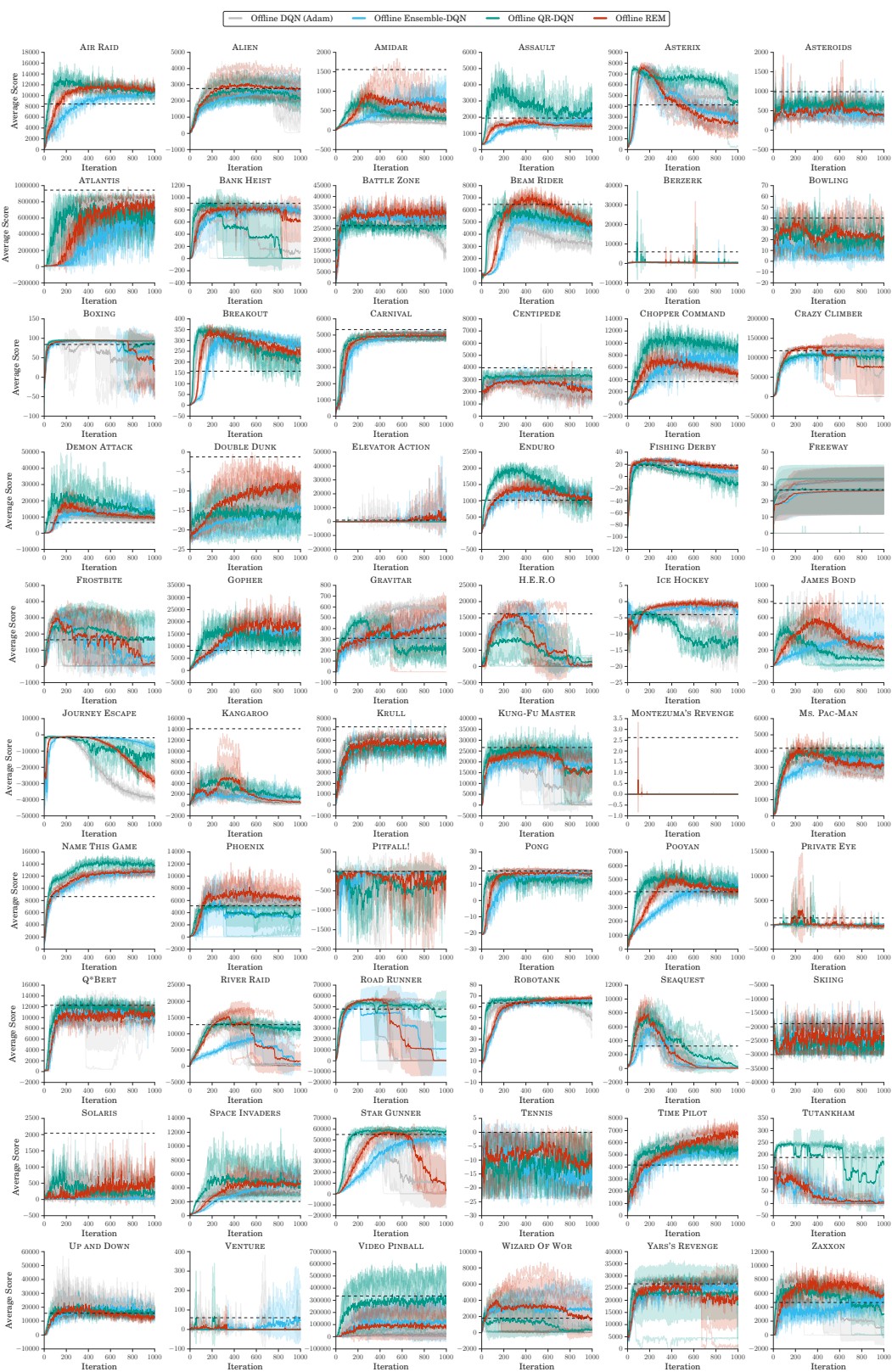

Figure A.7: Average evaluation scores across stochastic version of 60 Atari 2600 games of DQN (Adam), Ensemble-DQN, QR-DQN and REM agents trained offline using the DQN replay dataset. The horizontal line for online DQN show the best evaluation performance it obtains during training. All the offline agents except DQN use the same multi-head architecture with $K = 200$ heads. The scores are averaged over 5 runs (shown as traces) and smoothed over a sliding window of 5 iterations and error bands show standard deviation.

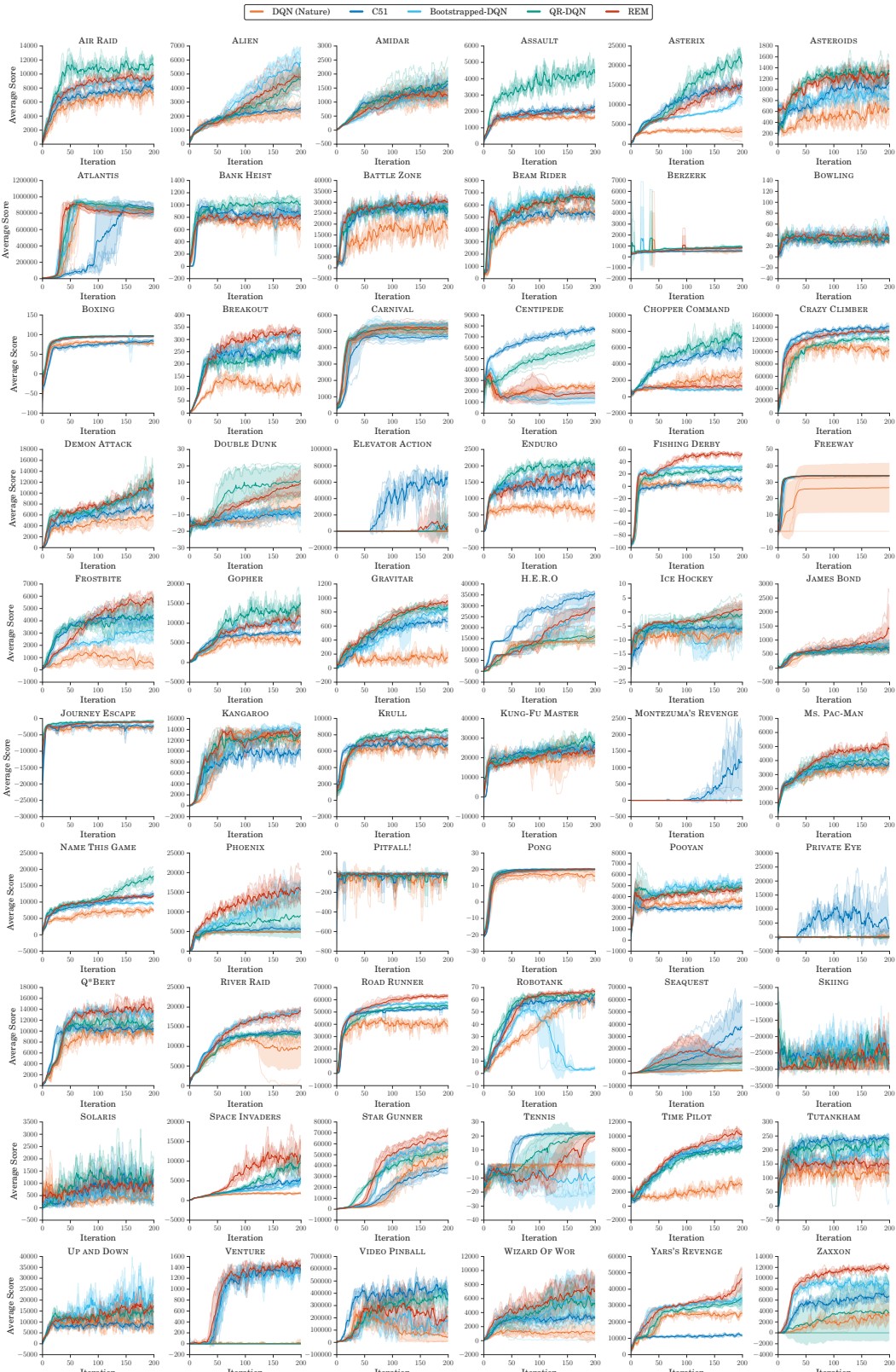

Figure A.8: **Online results.** Average evaluation scores across stochastic version of 60 Atari 2600 games of DQN, C51, QR-DQN, Bootstrapped-DQN and REM agents trained online for 200 million game frames (standard protocol). The scores are averaged over 5 runs (shown as traces) and smoothed over a sliding window of 5 iterations and error bands show standard deviation.

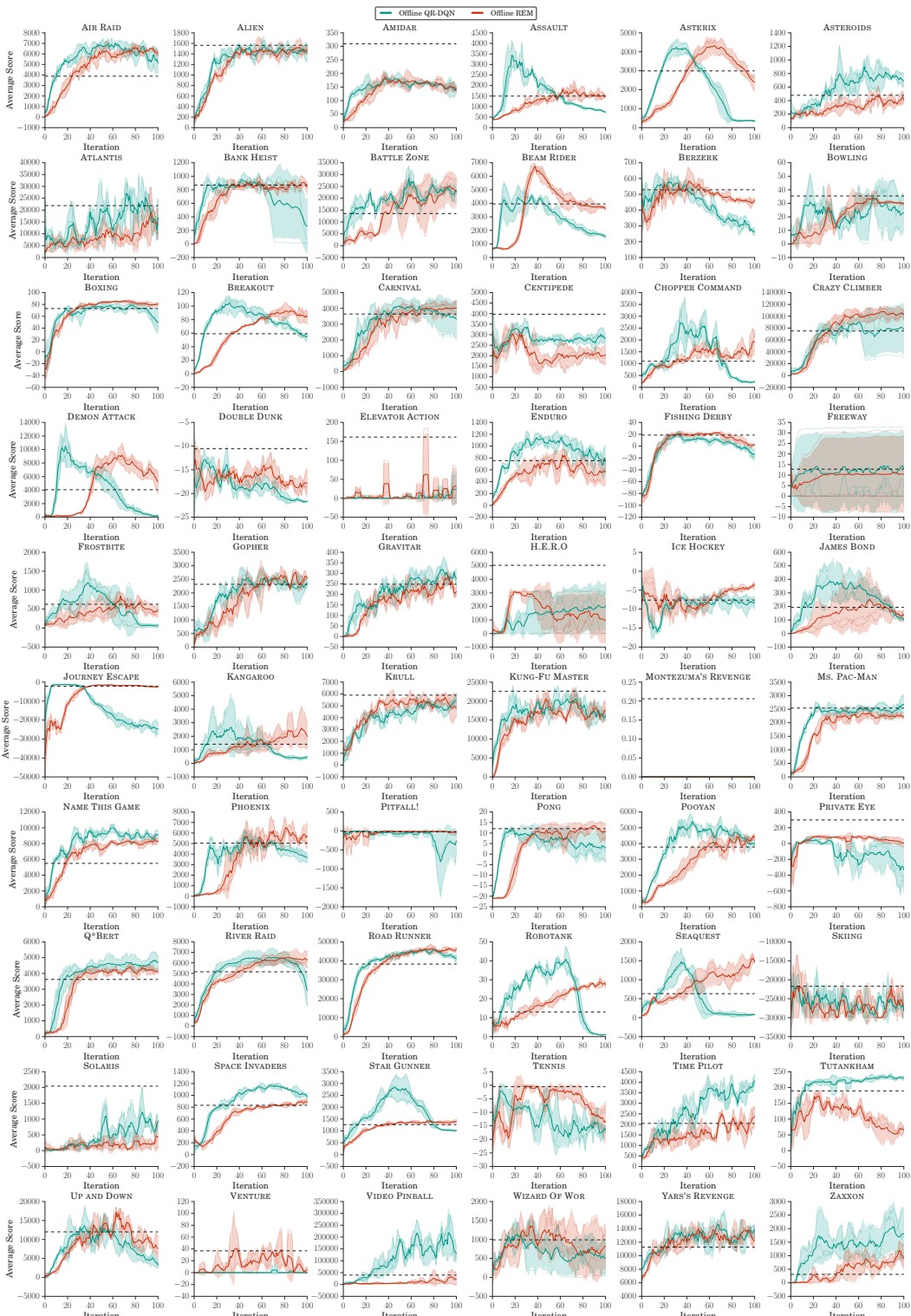

Figure A.9: **Effect of Dataset Quality.** Normalized scores (averaged over 3 runs) of QR-DQN and multi-head REM trained offline on stochastic version of 60 Atari 2600 games for 5X gradient steps using logged data from online DQN trained only for 20M frames (20 iterations). The horizontal line shows the performance of best policy found during DQN training for 20M frames which is significantly worse than fully-trained DQN.

