# OpenReview forum: "Striving for Simplicity in Off-Policy Deep Reinforcement Learning"
_ICLR.cc/2020/Conference — Reject_

### Official Review · AnonReviewer1 · 2019-10-23
**Official Blind Review #1**

**Rating:** 3

**Review:**

This paper investigated the offline settings of deep Q-networks, and provided relevant comparison among different agents. The authors also proposed two DQN variants by using ensembles, and showed the improvement (mainly from the random ensemble mixture) on Atari games, when compared with baselines.

In general, this paper explored an important problem, i.e., offline vs. online settings in DQNs. The experimental comparison also looked interesting, as it showed the empirical evidence on the influence of such settings. A lot of the experiments were performed to support the claims, which I really appreciate. On the other hand, it's a bit unclear to me whether offline should be preferred over its online counterpart, given the results presented and potential computational cost. For the proposed REM method, further explanation isolating from the offline vs. online setting could be helpful to understand why it works. Furthermore, some settings in the experiments need to be clarified. If the authors can address my concern, I would be willing to raise my score.

1. To my knowledge, using the logged experience to train an offline DQN agent is an interesting exploration. This approach can also achieve competitive performance, when compared with the online agent. My concern is whether the authors had considered the cost due to this offline setting, as additional time needs to be spent on training another agent to obtain such logged data, which is not required in the online methods. Moreover, Table 1 suggests that the online agents usually outperformed their offline counterparts, from which I am somewhat in doubt about whether the offline approaches should be preferred.
2. I am also wondering how the performance depends on the agent used to generate the logged data. In the current version, it is from the Nature DQN, but what will happen if we use a different (can be either better or worse) agent? For example, how about using C51 to generate the logged data?  I hope the authors can provide more evidence and analysis on why the offline settings should work.
3. When training the agents with the logged experience, how did you handle the new data generated? I understand the offline setting may not consider this issue, but am just thinking if such data can be better used.
4. The ensemble based DQNs proposed in Section 4 are very interesting, which can even be a separate contribution, as they do not depend on the online or offline settings. From this perspective, I am a bit curious if the authors have more explanation on the benefits of such methods, e.g., reducing the overestimation bias and gradient noise, as shown in the averaged-DQN and the following softmax-DQN work,
Song et al., Revisiting the Softmax Bellman Operator: New Benefits and New Perspective, ICML 2019,
which the authors may refer to in their paper.
5. For the curves in Figures 1 and 4, how about the corresponding running time? Such information could be helpful for others trying to reproduce the results.

**Experience Assessment:**

I have published in this field for several years.

**Review Assessment: Checking Correctness Of Derivations And Theory:**

I assessed the sensibility of the derivations and theory.

**Review Assessment: Checking Correctness Of Experiments:**

I carefully checked the experiments.

**Review Assessment: Thoroughness In Paper Reading:**

I read the paper thoroughly.

---

> ### Author Response · Authors · 2019-11-07
> **Response to Reviewer 1 continued..**
>
>
> > The ensemble based DQNs .. separate contribution.  ..  more explanation on the benefits of such methods ..
>
> Although the ensemble methods seem separate, their design was motivated by the search for a simpler algorithm with similar performance to offline QR-DQN using the same multi-head architecture. Additionally, the offline setup provides the insight that the gains from these methods are primarily due to exploitation. Our intuition behind REM was to consider Q-learning as a constraint satisfaction problem based on the Bellman optimality equation for each (s, a, r, s’) tuple and simply adding more constraints for regularization given access to multiple Q-value estimates.
>
> We believe that the effectiveness of REM is linked to the effectiveness of dropout in supervised learning. Empirically, Figure 4(a) shows that Ensemble-DQN (and Averaged-DQN) doesn’t show much improvement over DQN with Adam. However, since REM outperforms Averaged-DQN and DQN with Adam, we believe that the gains from REM are due to the noise from randomly ensembling Q-value estimates leading to more robust training analogous to dropout. Our experiments in Section 4.3 for REM with separate Q-network ensembles show improvements over REM with a shared multi-head Q-network further consolidates this belief as separate ensembles lead to more diverse individual Q-estimates leading to more robust training.
>
> >  curves in Figures 1 and 4, .. corresponding running time .. reproduce results
>
> Given the DQN replay dataset, the offline experiments are approximately 3X faster than the online experiments for the same number of gradient steps on a P100 GPU. In Figure 2 and 4(a), the offline agents are trained for 5X gradient steps, thus, the experiments are 1.67X slower than running online DQN for 200 million frames (standard protocol). Furthermore, since the offline experiments do not require any data generation, using tricks from supervised learning such as using much larger batch sizes than 32 with TPUs / multiple GPUS would lead to a significant speed up. We have included this in the uploaded revision in the experiment details section (A.4) in the appendix.
>
> References:
> 1. Scott Fujimoto, David Meger, and Doina Precup. Off-policy deep reinforcement learning without exploration. ICML, 2019.
> 2. Shangtong Zhang and Richard S. Sutton. A deeper look at experience replay. arXiv preprint arXiv:1712.01275, 2017.
> 3. Timothy P Lillicrap, Jonathan J Hunt, Alexander Pritzel, Nicolas Heess, Tom Erez, Yuval Tassa, David Silver,
> and Daan Wierstra. Continuous control with deep reinforcement learning. ArXiv:1509.02971, 2015.

---

> ### Author Response · Authors · 2019-11-07
> **Response to Reviewer 1**
>
> We thank the reviewer for their thoughtful comments and questions. We address them in order.
>
> > unclear .. whether offline should be preferred over its online counterpart ..
> We are not proposing offline RL as an alternative to online RL. Instead, we advocate that offline RL is an important problem for real-world RL that deserves more attention and provides a simpler and more reproducible setup than online RL to benchmark off-policy algorithms and tackle issues in off-policy RL isolation from exploration.
>
> Offline RL tries to exploit a fixed dataset as much as possible. Instead of competing with online RL, offline RL is potentially helpful for creating more sample efficient online algorithms as an ideal online agent should exploit the data collected so far during online training as much as possible before further collecting new data via exploration.
>
> > .. logged experience to train an offline DQN.. competitive performance .. cost due to this offline setting ..
>
> As mentioned above, we are not trying to compete with online RL. The offline optimization on logged DQN data is trying to answer whether it is possible to improve upon the best policy obtained training of online DQN by training completely offline on entire DQN replay of 200 million frames. This setup is motivated by the real-world RL problems where we already have access to (usually continually growing) diverse datasets of logged experiences and want to improve upon the existing best policy, e.g., data collected by hardcoded/random policies for robotics, historical data from recommender systems, sensory data logged from self-driving cars etc.
>
> > Moreover, Table 1 .. the online agents .. outperformed their offline counterparts, .. in doubt whether .. offline approaches should be preferred.
>
> Performance of online agents other than DQN is not comparable to offline agents trained on logged DQN data as the online agents collect their own different data. The comparison with online C51 in figures 1 and 4(a) is to show the magnitude of improvement from offline setup focusing only on exploitation, i.e., it is interesting that better optimization of DQN data can lead us to the same gains as an online C51 agent which is considered as a major improvement over DQN for off-policy deep RL.
>
> > .. how the performance depends on the agent used to generate the logged data. .. For example, using C51 to generate the logged data?
>
> In this work, we only claim that offline training on a large and diverse dataset with state-of-the-art RL agents (e.g. QR-DQN, TD3) works well in contrast to the findings by Fujimoto et al. (2019) and Zhang and Sutton (2017). In addition to Atari, our experiments on continuous control environments in Section 3.2 (also see Figure A.1 in the appendix) confirm this claim for data collected from an online DDPG agent (Lillicrap et al., 2015) with a highly stochastic policy. Furthermore, we expect offline training with standard RL algorithms to be successful as long as the fixed dataset is diverse (e.g. data collected from a broad mixture of policies) and large (see Section 6 for the effect of dataset size).
>
> We also ran experiments with entire replay data collected from an online QR-DQN agent and observe that the offline REM/QR-DQN still surpass gains from online C51 and significantly outperform the data-collecting policy (mixture of policies encountered during QR-DQN training).  Analogous to our results with DQN (Nature) data, we expect a better agent such as offline Rainbow to surpass online QR-DQN using logged QR-DQN data.
>
> > When training .. with logged experience, how did you handle the new data ..
>
> We didn’t generate any new data during the training of the offline agents with logged experience and only used online environments during evaluation of policies obtained during training. Although practical, adding new data generated during evaluation would complicate the offline setup and make it harder to compare exploitation ability of different agents. If the reviewer were referring to data generated during the online experiments in Figure 4(b), we'd like to clarify that no logged experiences were used for those experiments.

---

### Official Review · AnonReviewer2 · 2019-10-23
**Official Blind Review #2**

**Rating:** 3

**Review:**

*Summary*

Paper tackles an important issue, which is benchmarking off-policy deep RL algorithms. The motivation is important, because 1) assessing the performance of off-policy RL algorithm is hard due to the mixed effect of exploration and exploitation while collecting data; hence the paper suggest to fix the data collection process by a standard algorithm (DQN) and then compare the off-policy methods. 2) A fixed dataset will help the reproducibility of the results.

Additionally the paper assess the performance of some off-policy RL methods (mainly DQN and QR-DQN) and showed that given offline data, these algorithms can obtain a superhuman performance.

Besides the mentioned contribution, authors strived for a simple algorithm (easy to implement), and suggests a new simple algorithm, Random Ensemble Mixture (REM) by training multiple Q-functions and randomly sampling a subset for ensemble estimate. The mina motivation come from supervised learning literature.

*Decision*

I enjoyed reading the paper and I think authors tackled a very important problem. Although a very important step, I vote for a weak reject for this paper as I believe the contribution of the current paper is limited. I made my decision mainly based on the following points:

[Limited Task Diversity]: I agree with the authors that having a fix offline data for testing different algorithms is important; however the suggestion (DQN logged data on Atari2600) is a very limited dataset, in terms of 1) task diversity 2) Data collection strategy.
For a benchmark, I would like the dataset to have different strategies for data collection, so that the effect of data collection can be stripped off. In the current assessment, we compare the performance of algorithms condition on the DQN e-greedy data collection procedure. (For example, the right claim to make is: QR-DQN showed a better performance condition on the current data generation process). And it is not clear to me the same observation will hold given a different data generation process (for example a random policy, or a human playing data, or …)
Also, I think only focusing on Atari tasks is limited in nature, as many of these games share a similar underlying mechanism.


[Overfitting] I believe having a fixed dataset for testing off-policy RL is great, however the current suggestion is very prone to overfitting. I can foresee that if the community start to use this benchmark as a test case, soon we will overfit to this task, in a sense that I try algorithm X on offline data, test on Atari see the results, tune algorithm X test on Atari again, … This way I finally have an algorithm that only learned from offline data, but is performing great on online Atari. But I am overfitting!


[Simplicity Trade-Off] Authors focused on getting a simple algorithm, but to me it is unclear why should we optimize for simple algorithm, and my impression of simple as authors have in mind is a “easy to implement” algorithm.
Simplicity is great, but usually I believe we seek a simple algorithm to be able to get more insight into the theory and build more intuition. However, in this work there is a lot emphasis on simple algorithm, but no extra significant insight into off-policy learning has been gained. Additionally, I found the suggested algorithm ad-hoc and of limited contribution.


Minor Comments/ Question that did not significantly contribute to my decision:

[Valid Q-function]: authors mentioned a convex combination of Q-values are a valid Q-value, but never defined a “valid” Q-value. Do authors mean, if Q1 is generated by \pi_1 and Q2 by \pi_2 then convex combination of Q1 and Q2, Q^* can be also generated by some policy \pi^*?
If so, this needs clarification.
And if so, I’m not sure why this is important, since in Q-learning the Q-values that we obtain before convergence might be “invalid” (by the definition above, so that no real policy can generate those values exactly), so what is the point of paying attention to a valid Q values in the first place?

*Improvement*

I think some clear improvement avenues for the future are
1. More diverse set of tasks: I would like to see some non-Atari tasks included in the dataset.
2. More diverse data collection strategy: Example can be different exploration strategies than e-greedy. Or even random exploration, or maybe data of human players. Since we don’t want to have an algorithm that is good only for state distribution induced by e-greedy exploration.
3. More diverse task will help with overfitting by itself. But a good benchmark needs a test case that cannot be exploited.
4. More insight into why REM is outperforming QR-DQN or other algorithms. Currently I am not convinced that we understand why REM is outperforming, or if it’s only because we designed it to be good for this specific task and dataset?


Generally, I believe authors made a very important step, and I really enjoyed reading the paper; however, I think the current contribution is not sufficient to merit a publication. This makes a good workshop paper at this point.


**Experience Assessment:**

I have read many papers in this area.

**Review Assessment: Checking Correctness Of Derivations And Theory:**

I assessed the sensibility of the derivations and theory.

**Review Assessment: Checking Correctness Of Experiments:**

I assessed the sensibility of the experiments.

**Review Assessment: Thoroughness In Paper Reading:**

I read the paper at least twice and used my best judgement in assessing the paper.

---

> ### Author Response · Authors · 2019-11-07
> **Response to Reviewer 2 continued..**
>
> > [Overfitting]  continued ..
>
> In supervised learning, we never access the test set during training and use validation performance as a proxy for test performance. However, in RL, we don’t have access to a good proxy yet for online performance given only offline data (this is the well known problem of *off-policy policy evaluation*), and therefore we use online evaluation as the performance metric. However, we ensure that the online evaluation is done only for a limited number of times to avoid overfitting (e.g., we evaluate each offline agent for a total of 1000 times). With development of better off-policy policy evaluation metrics in the future, we hope that both the training and evaluation can be *completely offline*.
>
> > [Simplicity Trade-Off]:
>
> >> "REM only “easy to implement” .. suggested algorithm ad-hoc and of limited contribution."
>
> REM was inspired by using the same architecture changes as QR-DQN while still doing expected Q-learning which minimizes the scalar TD-error. It is algorithmically simpler than QR-DQN which minimizes a probability distance measure from a distributional Bellman target. Furthermore, the simplicity of REM is evident by the fact that it converges to Q* in the tabular setting while QR-DQN may not.
>
> >> ".. REM designed to be good for this specific task and dataset?"
>
> We show the competence of REM in both the offline setting with DQN replay datasets as well as the online setting (Figure 4(b)). We assert that developing an off-policy algorithm that works well across all the 60 Atari games is not easy. Moreover, most prior deep RL research with discrete actions has used Atari as the sole benchmark.
>
> >> ".. no extra significant insight into off-policy learning has been gained."
>
> REM provides the insight that Q-learning can be thought of as a constraint satisfaction problem based on the Bellman optimality equation for each (s, a, r, s’) tuple and REM regularizes Q-learning by adding more consistency constraints given access to multiple Q-value estimates.
>
> >> "More insight into why REM is outperforming QR-DQN or other algorithms .."
>
> We believe that the effectiveness of REM is linked to the effectiveness of dropout in supervised learning. Empirically, since REM outperforms Ensemble-DQN and DQN (Adam) (see Figure 4(a)), we believe that the gains from REM are due to the noise from randomly ensembling Q-value estimates leading to more robust training analogous to dropout. Our experiments in Section 4.3 for REM with separate Q-network ensembles show improvements over REM with a shared multi-head Q-network further consolidates this belief as separate ensembles lead to more diverse individual Q-estimates leading to more robust training.
>
> > Minor Clarifications:
>
> >> ".. the paper assess the performance of some off-policy RL methods (mainly DQN and QR-DQN) and showed that given offline data, these algorithms can obtain a superhuman performance."
>
> Offline DQN(Nature) underperforms the online DQN(Nature) on most games and doesn’t achieve superhuman performance while offline QR-DQN outperforms online DQN.
>
> >> "REM by training multiple Q-functions and randomly sampling a subset for ensemble estimate."
>
> We don’t randomly subsample a subset but use a random convex combination of all the Q-value estimates.
>
> >> [Valid Q-function]: "authors mentioned a convex combination of Q-values are a valid Q-value, but never defined a “valid” Q-value."
>
> We mention that a convex combination is a valid *estimate* of the Q-value (i.e., it’s still an estimate of the optimal Q-values as opposed to some random combination, e.g., 0.2 * Q1 + 0.5 * Q2). As pointed by the reviewer, the combination may or may not be a valid Q-value. To alleviate confusion,  we have revised the paper to call the convex combination a Q-estimate rather than a valid Q-estimate.
>
> References:
> 1. Scott Fujimoto, David Meger, and Doina Precup. Off-policy deep reinforcement learning without exploration. ICML, 2019.
> 2. Shangtong Zhang and Richard S. Sutton. A deeper look at experience replay. arXiv preprint arXiv:1712.01275, 2017.
> 3. Timothy P Lillicrap, Jonathan J Hunt, Alexander Pritzel, Nicolas Heess, Tom Erez, Yuval Tassa, David Silver,
> and Daan Wierstra. Continuous control with deep reinforcement learning. ArXiv:1509.02971, 2015.
> 4. Aviral Kumar, Justin Fu, George Tucker, and Sergey Levine. Stabilizing Off-Policy Q-Learning via Bootstrapping Error Reduction. NeurIPS, 2019.
> 5. Anonymous. Behavior regularized offline reinforcement learning. 2020. URL https://openreview.net/forum?id=BJg9hTNKPH . under review

---

> ### Author Response · Authors · 2019-11-07
> **Response to Reviewer 2**
>
> We thank the reviewer for their thoughtful comments and suggested improvements. We address their concerns below.
>
> > [Limited Task Diversity]:
>
> >> "More diverse set of tasks: I would like to see some non-Atari tasks included in the dataset"
>
> The Atari benchmark is the most popular and commonly used discrete RL benchmark and the entire set of 60 different games makes it hard to overfit an RL algorithm to a specific game. Recent work (Fujimoto et al., 2019, Kumar et al., 2019, Anonymous, 2020) has only tackled offline RL on continuous control using MuJoCo gym environments.  Based on the popularity and ease of experimentation on Atari, we hope that an *appropriate subsampling* of the entire DQN replay dataset would spur offline RL research without having to deal with continuous actions.
>
> Moreover, the diversity of the benchmark depends on the problems it is used for tackling; we assert that the logged datasets across 60 different Atari games provide a sufficiently diverse benchmark to tackle the issues of sample efficiency or stability in off-policy RL (please see Section 6).
>
> >> More Diverse Data collection strategy ..
>
> >> "performance of algorithms condition on DQN e-greedy data collection procedure .. same observation .. given a different data generation process"
>
> We'd like to emphasize that the collected data contains samples from all of the intermediate policies seen during the optimization of the DQN (Nature) agent, i.e., the behaviour policy is a huge mixture of policies and is non-markovian and can’t be induced by a single ε-greedy policy.
>
> We only claim that offline training on a large and diverse dataset with state-of-the-art RL agents (e.g. QR-DQN, TD3) works well in contrast to the findings by Fujimoto et al. (2019) and Zhang and Sutton (2017). In addition to Atari, our experiments on continuous control environments in Section 3.2 (also see Figure A.1 in the appendix) confirm this claim for data collected from an online DDPG agent (Lillicrap et al., 2015) with a highly stochastic policy. We expect offline training with standard RL algorithms to be successful as long as the fixed dataset is diverse (e.g. data collected from a broad mixture of policies) and large (see Section 6 for the effect of dataset size).
>
> >> "Different exploration strategies: Or even random exploration, or maybe data of human players"
>
> In this work, we mainly experimented with using the entire replay dataset to approximately replicate the real-world scenario with large diverse datasets (e.g. historical data from recommender systems, data collected by hardcoded / random / existing policies for robotics) and confirm that recent deep RL algorithms perform well with diverse offline datasets as opposed to the prior findings.
>
> Although developing an offline RL algorithm that works with different data collection schemes was not the main motivation in our paper,  we agree that using different data collection schemes would improve the utility of the benchmark.  Since the DQN replay dataset for each game is composed of 5 runs of 200 million frames stored in an ordered fashion as they were observed during online DQN training (although we don’t use the ordering in this work), one possible way to create different data collection schemes is to take the first k million frames to provide data close to random (data with low returns) while using only the last k million frames is analogous to using data provided by an expert.
>
> We have launched new offline experiments across all 60 Atari games using the first 20 million frames and on 6 games using 40 million frames (since these settings emulate the realistic scenario of offline Q-learning from initial exploration data) and will upload a revision with those results before the end of the discussion period.
>
> > [Overfitting]  "current suggestion is prone to overfitting ..  a good benchmark needs a test case that cannot be exploited."
>
> We assert that the current proposal is less prone to overfitting than the prevalent online RL setting where the training and evaluation are both done using the online Atari environments, where the agent can access the online environment as many times as it would like (although we limit the interaction to 200 million frames in practice). Furthermore, hyperparameter tuning on Atari is usually done on a small number of games (5 or 6) and then the hyperparameters found using these small number of games are used across all the 60 games which also reduces the chances of overfitting.

---

### Official Review · AnonReviewer3 · 2019-10-24
**Official Blind Review #3**

**Rating:** 3

**Review:**

#rebuttal response
Thanks for your response and the additional experiments. I do think it is a very interesting paper but I keep my score. First, the improvement of their model is not very significant both in Table 1 and the standard RL setting. Second, I am still not clear about how off-policy learning benefits the study on standard RL. Since exploration and exploitation interact with each other, disentanglement between them cannot reflect the original difficulty of RL problems. Different exploration strategies or balancing methods between exploration and exploitation will produce different data. I guess this is why the data generation process is a common concern raised by both Reviewers 1 and 2. I think the response of the authors is not enough to clarify this. In addition, I cannot find Figures 4a and 4b. I think this paper is not ready for publication right now but I encourage the authors to improve this work and resubmit it in the future.


#review

This paper collects a replay dataset of Atari to propose a benchmark for offline (batch) reinforcement learning. Experiments on this dataset show that offline RL algorithms can outperform online DQN. Then, this paper presents a new off-policy RL algorithm, which uses random convex combinations of multiple networks as the Q-value estimator. The experimental results show that it outperforms distributional RL algorithms (i.e., C51 and QR-DQN) in the offline setting and performs comparably in the online setting.

The idea of randomly combining Q networks is interesting and paper is well written to demonstrate their key ideas. However, I have some concerns.

Why are off-line RL algorithms necessary? I am not sure this paper is relevant to the community of standard RL. This paper does not show the usage of their method to a standard RL agent. I am happy to change my score if the authors can clarify the motivation of studying the offline performance of conventional RL algorithms and clearly show the improvement over baseline algorithms in a standard RL setting rather than only show insights for potential usage in online RL.

In my opinion, the results are not significant to support their claims. Such a small gap (4.9% as shown in Table 1) over baseline models may result from considerable hyper-parameter tunning. In addition, the figures and tables in this paper do not show deviations and confidence intervals.

Since REM has similar architectures with attention networks, it will be better to include an attention-based Q network as a baseline model.

In the right sub-table of Table 1, I wonder why the performance of online QR-DQN in this paper is much lower than that reported in the original papers (media is about 200%).



**Experience Assessment:**

I have published one or two papers in this area.

**Review Assessment: Checking Correctness Of Derivations And Theory:**

I assessed the sensibility of the derivations and theory.

**Review Assessment: Checking Correctness Of Experiments:**

I carefully checked the experiments.

**Review Assessment: Thoroughness In Paper Reading:**

I read the paper at least twice and used my best judgement in assessing the paper.

---

> ### Author Response · Authors · 2019-11-06
> **Response to Reviewer 3 continued ..**
>
> > “Since REM has similar architectures with attention networks, it will be better to include an attention-based Q network as a baseline model.”
>
> REM has the same multi-head architecture as QR-DQN and is simply using an ensemble of Q-value estimates using random mixing proportions. The attention-based Q-network would involve optimizing the mixing proportions of different Q-value heads, which is equivalent to learning a single Q-value estimate (i.e. training DQN using the multi-head REM architecture followed by an additional linear layer). Our preliminary experiments suggests that this architecture is not effective and the benefits of REM stem from randomness of the mixing proportions and not the architecture (also see Figure 4(a)). By analogy, the benefits of dropout stem from randomness of the dropout masks and such benefits go away if one learns the dropout masks. We are happy to include a comparison in the final version if the reviewers find it useful.
>
> References:
> 1. Gabriel Dulac-Arnold, Daniel Mankowitz, and Todd Hester. Challenges of real-world reinforcement learning. arXiv preprint arXiv:1904.12901, 2019.
> 2. Pablo Samuel Castro, Subhodeep Moitra, Carles Gelada, Saurabh Kumar, and Marc G Bellemare. Dopamine: A research framework for deep reinforcement learning. ArXiv:1812.06110, 2018.
> 3. Marc G Bellemare, Will Dabney, and Rémi Munos. A distributional perspective on reinforcement learning. ICML, 2017.
> 4. Marlos C Machado, Marc G Bellemare, Erik Talvitie, Joel Veness, Matthew Hausknecht, and Michael Bowling. Revisiting the arcade learning environment: Evaluation protocols and open problems for general agents. Journal of Artificial Intelligence Research, 2018.
> 5. Scott Fujimoto, David Meger, and Doina Precup. Off-policy deep reinforcement learning without exploration. ICML, 2019.
> 6. Shangtong Zhang and Richard S. Sutton. A deeper look at experience replay. arXiv preprint arXiv:1712.01275, 2017.

---

> ### Author Response · Authors · 2019-11-11
> **Response to Reviewer 3**
>
> > “Why are off-line RL algorithms necessary?”
>
> In many real-world applications of RL such as digital advertising, recommender systems, and healthcare, one cannot deploy a learning agent into production due to the complex requirements of the production setting, but one can learn from a vast amount of offline logged data available. When discussing challenges of real-world RL, Dulac-Arnold et al. (2019) conclude that offline RL is the only feasible solution for many real-world problems, which do not have a simulator or when a simulator is too complex to build. In addition, offline RL focuses on the exploitation aspect of the RL problem and provides us with a simpler and more reproducible experimental setup to study important problems such as sample efficiency and stability of RL algorithms in isolation (without exploration -- see Section 6).
>
> > “ … clarify the motivation of studying the offline performance of conventional RL algorithms”
>
> The paper aims to (1) isolate the contributions of exploitation vs. exploration in off-policy deep RL algorithms, (2) help improve reproducibility of deep RL research, and (3) facilitate the design of simpler and better deep RL algorithms.
>
> We investigate the common wisdom in the deep RL community (e.g., Fujimoto et al., 2019) that standard RL algorithms fail in the offline RL setting and the claim that online interactions are crucial for Q-learning methods (Zhang and Sutton, 2017). Our results in Section 3 on Atari and MuJoCo environments contradict prior findings and show that offline training on large and diverse replay datasets using state-of-the-art RL agent (e.g., QR-DQN, TD3) performs surprisingly well.
>
> > “ … clearly show the improvement over baseline algorithms in a standard RL setting”
>
> Section 4.4 and Figure 4(b) compare different online agents in the standard RL setting, where we demonstrate that online REM compares favorably with online QR-DQN, a state-of-the-art distributional agent on Atari games. We will make this section more visible in the final version of the paper.
>
> > “I wonder why the performance of online QR-DQN in this paper is much lower than that reported in the original papers”
>
> The median improvement reported in the original papers is based on deterministic Atari games. We report the results on the stochastic version of Atari games with sticky actions, the standard protocol for Atari experiments, as recommended by Machado et. al (2018). We have updated the text and all the captions in the new version to mention the use of stochastic Atari more explicitly.
>
> For example, C51 (Section 5.2, Bellemare et al., 2017) paper reports an improvement of 178% in median normalized scores on deterministic Atari, while a median improvement of only 21.5% on stochastic Atari with 3 runs per game. We use 5 runs per game and use the hyperparameters provided in Dopamine (Castro et. al, 2018) baselines for a standardized comparison and observe a 19% median improvement for C51.
>
> > “Such a small gap (4.9% as shown in Table 1) over baseline models may result from considerable hyper-parameter tunning.”
>
> In the offline setting, QR-DQN outperforms averaged Ensemble-DQN by 6.8% and we outperform QR-DQN by 4.9% in terms of median normalized scores which are calculated based on 60 Atari games and 5 runs per game. Given the simplicity of REM over QR-DQN and the fact that these results are on stochastic Atari, a harder benchmark than deterministic Atari, these results are significant. . We did not do any hyperparameter tuning for offline REM and simply used the parameters provided by Dopamine baselines. Please refer to section A.4 in the appendix for more details.
>
> > In addition, the figures and tables in this paper do not show deviations and confidence intervals.
>
> All of the learning curves in the appendix (Figure A.4 - A.8) show the standard deviations. Median normalized scores reported in figures and table 1 are the median across all 60 games of scores averaged over 5 runs per game and the calculation of standard deviation of median normalized scores is extremely expensive. The standard deviation for these scores requires rerunning all of the offline experiments multiple times to get multiple median estimates, and we are not aware of any prior work reporting it.  Instead, we have revised the paper to add individual runs as traces in Figure 1 and 4 to show the variations in our average results.

---

### Author Response · Authors · 2019-11-15
**Revision with new experiments and clarifications posted**

Based on reviewer feedback, we have posted a revision with the following changes:

1. Per R3's feedback, we have (1) improved the visibility of the online RL results, (2) more explicitly mention the use of stochastic version of Atari in captions to avoid confusion with deterministic Atari results, (3) clarified our motivation for studying offline RL, and (4) added individual runs as traces in figures 1 and 4(a) to show variation in our results.

2. Per R2's feedback, we have added a discussion (Section 6) for specifying ways to generate various data collection strategies for benchmarking offline RL and mention problems (e.g. sample-efficiency, stability) which can be tackled in isolation from exploration using the offline benchmark. We also added new offline experiments on all Atari games in Section 5 using only the first 20 millions frames from the entire DQN replay dataset to mimic offline RL setting using exploration data with suboptimal returns. We have also updated the abstract to highlight our contributions in addition to the benchmark.

3. Per R1's feedback, we have (1) included the wall clock time information for the offline experiments, (2) added our hypothesis regarding why REM outperforms existing algorithms and our intuition behind REM, (3) improved Section 5 which contains ablation experiments showing the effect of dataset size and composition on our offline RL results and (4) clarified the utility of the offline RL setting.

We also kindly refer the reviewers to related work section which shows the contrast in our findings with respect to existing work in offline deep RL.  It is unfortunate that the reviewers were not able to engage in discussion with us but we hope that the revision and author response address their concerns.

---

### Author Response · Authors · 2020-09-23
**Accepted at ICML 2020 with updated title**

An Optimistic Perspective on Offline Reinforcement Learning -- https://offline-rl.github.io

---

### Decision · Program_Chairs · 2019-12-19

**Decision:**

Reject

**Comment:**

All the reviewers recommend rejecting the submission. There is no basis for acceptance.